# Preexisting risk-avoidance and enhanced alcohol relief are driven by imbalance of the striatal dopamine receptors in mice

Miriam E. Bocarsly [1,2,8] ✉, Marlisa J. Shaw[1,3], Emilya Ventriglia[4], Lucy G. Anderson[5], Hannah C. Goldbach [1,5], Catherine E. Teresi[1,6], Marilyn Bravo[1], Roland Bock [1,5], Patrick Hong[2], Han Bin Kwon [1], Imran M. Khawaja[2], Rishi Raman [2], Erin M. Murray [1], Jordi Bonaventura [4,7], Dennis A. Burke [1], Michael Michaelides [4] & Veronica A. Alvarez [1,4,5,6,8] ✉

Alcohol use disorder (AUD) is frequently comorbid with anxiety disorders, yet whether alcohol abuse precedes or follows the expression of anxiety remains unclear. Rodents offer control over the first drink, an advantage when testing the causal link between anxiety and AUD. Here, we utilized a risk-avoidance task to determine anxiety-like behaviors before and after alcohol exposure. We found that alcohol's anxiolytic efficacy varied among inbred mice and mice with high risk-avoidance showed heightened alcohol relief. While dopamine D1 receptors in the striatum are required for alcohol's relief, their levels alone were not correlated with relief. Rather, the ratio between striatal D1 and D2 receptors was a determinant factor for risk-avoidance and alcohol relief. We show that increasing striatal D1 to D2 receptor ratio was sufficient to promote risk-avoidance and enhance alcohol relief, even at initial exposure. Mice with high D1 to D2 receptor ratio were more prone to continue drinking despite adverse effects, a hallmark of AUD. These findings suggest that an anxiety phenotype may be a predisposing factor for AUD.

Alcohol use disorder (AUD) and anxiety are among the most commonly diagnosed psychiatric disorders in clinical populations and have a high rate of comorbidity[1–3]. Individuals suffering from alcohol use disorder are 2.6 times more likely to meet diagnostic criteria for anxiety disorders (12 month-odds[4]). While the epidemiological data undoubtedly supports comorbidity, whether the anxiety phenotype precedes or is a consequence of alcohol misuse continues to be a topic of debate[5].

Some argue that an anxiety phenotype may precede the onset of alcohol drinking. In both humans and rodents, alcohol is a potent anxiolytic, which could explain why individuals suffering from mood and anxiety disorders use alcohol to cope with the difficult symptoms associated with these disorders[6]. In support of this "self-medication" hypothesis, epidemiological studies using cross-sectional and longitudinal data suggest that anxiety precedes the onset of problematic drinking[7–9]. The other side of the debate suggests that the anxiety phenotype results from withdrawal from chronic, heavy alcohol drinking[1,10,11]. In humans, signs of withdrawal include anxiety, depression, and increased sensitivity to stressors[12–14], which are positively correlated with alcohol relapse rates[15–17]. Similarly, some preclinical

[1]Laboratory on Neurobiology of Compulsive Behaviors, National Institute on Alcohol Abuse and Alcoholism, Intramural Research Program, NIH, Bethesda, MD, USA. [2]Department of Pharmacology, Physiology and Neuroscience, Brain Health Institute, Rutgers New Jersey Medical School, Newark, NJ, USA. [3]NIH Academy Enrichment Program, Office of OITE, NIH, Bethesda, MD, USA. [4]National Institute on Drug Abuse, Intramural Research Program, NIH, Baltimore, MD, USA. [5]National Institute on Mental Health, NIH, Bethesda, MD, USA. [6]Center on Compulsive Behaviors, NIH, Bethesda, MD, USA. [7]Institut de Neurociències, Universitat de Barcelona Barcelona, Spain. [8]These authors jointly supervised this work: Miriam Bocarsly, Veronica A. Alvarez.
✉ e-mail: bocarsme@njms.rutgers.edu; alvarezva@mail.nih.gov

studies suggest that rodents abstaining after prolonged alcohol exposure show increased anxiety-like behavior[18]. In these studies, rodents in abstinence from chronic alcohol exposure display higher risk-aversion phenotype and, depending on the duration and level of chronic exposure, rodents in withdrawal often show higher alcohol consumption[19,20], suggesting that these affective withdrawal symptoms might be an important factor in driving relapse[20,21]. This observation supports the hypothesis that patients suffer from anxiety as a consequence of long-term exposure to alcohol[22]. However, the topic is still debated, and other rodent studies show mood disturbances during abstinence, such more immobility in forced swim test and sucrose anhedonia, without changes in risk-avoidance and anxiety-like behaviors[1,23].

Disentangling the relationship between anxiety and AUD will clarify our understanding of the risk factors of each disorder and inform treatments. Preclinical work in rodents is ideally suited for this goal because it allows control over the timing of first alcohol exposure and the ability to measure baseline expression of anxiety behavior and the anxiolytic effects of alcohol. Further, preclinical studies hold promise for developing a deeper understanding of the neural mechanisms underlying both alcohol abuse and anxiety[1,24].

AUD is a complex disorder characterized by dysfunction across multiple brain regions and circuits[25,26]. These brain regions are also affected in mood disorders like anxiety. Similarly, in rodents, acute and chronic alcohol exposure have been shown to alter the activity and the connectivity of neurons in basal ganglia, amygdala, bed nucleus of the stria terminalis and cortex, among others[18,27–29]. Recently, the dorsomedial striatum, an area of the basal ganglia implicated in cognitive flexibility, has received significant attention with multiple studies showing it is the target of alcohol-induced synaptic plasticity and circuit and behavioral adaptations[30–34]. Our previous work also has shown that dopamine D1 receptors in this region mediate alcohol-induced stimulation[35]. This complements findings from others showing that D1-receptor expressing striatal neurons also regulate alcohol seeking and consumption[33].

Here, we explored the hypothesis that a robust anxiolytic response to alcohol enhances its reinforcing properties, rendering some individuals more susceptible to developing AUD associated behaviors. Indeed, we found that, in mice, a preexisting risk-avoidance phenotype is associated with enhanced alcohol anxiety relief and increased drinking despite adverse consequences. The underlying mechanism involves a shift in the balance of dopamine D1- and D2-receptor expression in the striatum. The results of this study support the self-medication hypothesis and suggest that both an anxiety phenotype and high ratio of striatal D1 to D2 receptors, precede and are risk factors for AUD. At the same time, these results do not necessarily rule out the alternative hypothesis that withdrawal from chronic, heavy alcohol drinking can promote or exacerbate anxiety and mood disorders. To the contrary, these findings maintain the potential that a pre-existing anxiety phenotype and alcohol-induced mood disturbances could synergize, fueling the cycle of addiction.

## Results

### Individual variability in exploration-avoidance behavior in inbred C57BL/6J mice

We evaluated exploration-avoidance behavior in male and female C57BL/6J mice through three well-established tasks: the elevated zero maze (EZM), light-dark box, and open-field test[36]. These tasks, aimed at evaluating anxiety-like behaviors in rodents, create conflict between the drive to explore the novel environment and the drive for safety, avoiding exposed, bright zones. To capitalize on the novelty of the testing arena, each task was performed once per animal, employing a Latin square design. We observed variability in baseline levels of risk-avoidance behavior within this inbred mouse strain (Fig. 1A–C, Supplemental Fig. 1A, B). Within subjects correlations were found across

tasks; mice with increased avoidance on the EZM also tended to avoid the light compartment of the light-dark box (Supplemental Fig. 1C–G). These findings suggest a potential biological foundation for the observed variability, emphasizing the importance of individualized assessment.

### Individual variability in alcohol anxiolytic potency correlates with consumption in mice

C57BL/6J mice that received alcohol (1.2 g/kg, i.p.) showed an overall increase in exploratory behavior on both the EZM and the light-dark box, compared to those receiving saline (Fig. 1A–C). This difference was used to assess alcohol's anxiolytic properties and the dose was chosen based on previous reports indicating that 1.2 g/kg is sufficient to produce robust alcohol effects while minimizing the number of mice excluded due to falling off the maze[37,38]. In the EZM, mice exposed to alcohol spent less time in the open zones ($38 \pm 2\%$ after saline and $46 \pm 2\%$ after alcohol; Fig. 1B). In the light-dark box, mice showed increased exploration of the light zone ($36 \pm 1\%$ after saline and $41 \pm 1\%$ after alcohol; Fig. 1C). A positive correlation was observed between the time spent exploring the EZM open zones and the light compartment of light-dark box following alcohol administration (Supplemental Fig. 1H–L). The alcohol dose employed here (1.2 g/kg) does not increase overall locomotor activity, arguing against a contribution from the stimulatory effects of alcohol which are observed at higher doses (2 g/kg; Fig. 1D). To control for the development of tolerance with repeated alcohol exposures, we compared EZM performance of mice without previous alcohol exposure to those that had repeated alcohol exposure in the home cage and saw no differences in time spent in open zones (Supplemental Fig. 2A).

Another group of mice were tested in the EZM after alcohol and later given access to both water and 20% alcohol to drink in a two-bottle choice test (Fig. 1E). Mice that spent more time in the open zones following alcohol went on to consume more alcohol (Fig. 1F), providing evidence for a link between the effects of alcohol on risk-avoidance and increased voluntary consumption. No correlation was found between performance on the EZM and water consumption (Fig. 1G) or total fluid intake (Supplementary Fig. 2B), suggesting that this effect is specific to alcohol and not polydipsia in general.

Interestingly, alcohol produced no discernible change in the mean time spent in the center of the open field (Supplementary Fig. 1A, B), in agreement with previous reports[39]. Nor was time spent in the center of the open field following alcohol correlated with the time spent in the open zones of the EZM or light-dark box (Supplemental Fig. 1H–L).

### Alcohol anxiolytic effects are preferentially observed in mice with high risk-avoidance

We set out to investigate the relationship between risk-avoidance behavior and the potency of alcohol relief from this phenotype, using a within subject's design. Here, we employed repeated EZM tests in the same mice. While there is some controversy in the field, the stability of EZM performance across trials had been previously demonstrated in C57BL/6J mice[40] and we further validated it in our laboratory. Mice assessed on the EZM twice after saline administration with tests conducted one week apart showed similar time in the open zones across the two trials (Fig. 1H–I), and no sex differences were found (Supplemental Fig. 2K).

A separate cohort of mice underwent repeated EZM measures a week apart after both saline and alcohol, in a randomized order (Fig. 1J). Consistent with findings of Fig. 1A, B, this within-subjects experiment confirmed that alcohol increased overall exploration of the open zones (Fig. 1K). No differences were noted between male and female mice and no pervasive order effects of treatment were noted (Supplemental Fig. 2C–F). To parse out the effects of baseline risk avoidance, mice were categorized into "high" or "low" risk-avoidance groups based on their EZM performance after saline. Alcohol increased

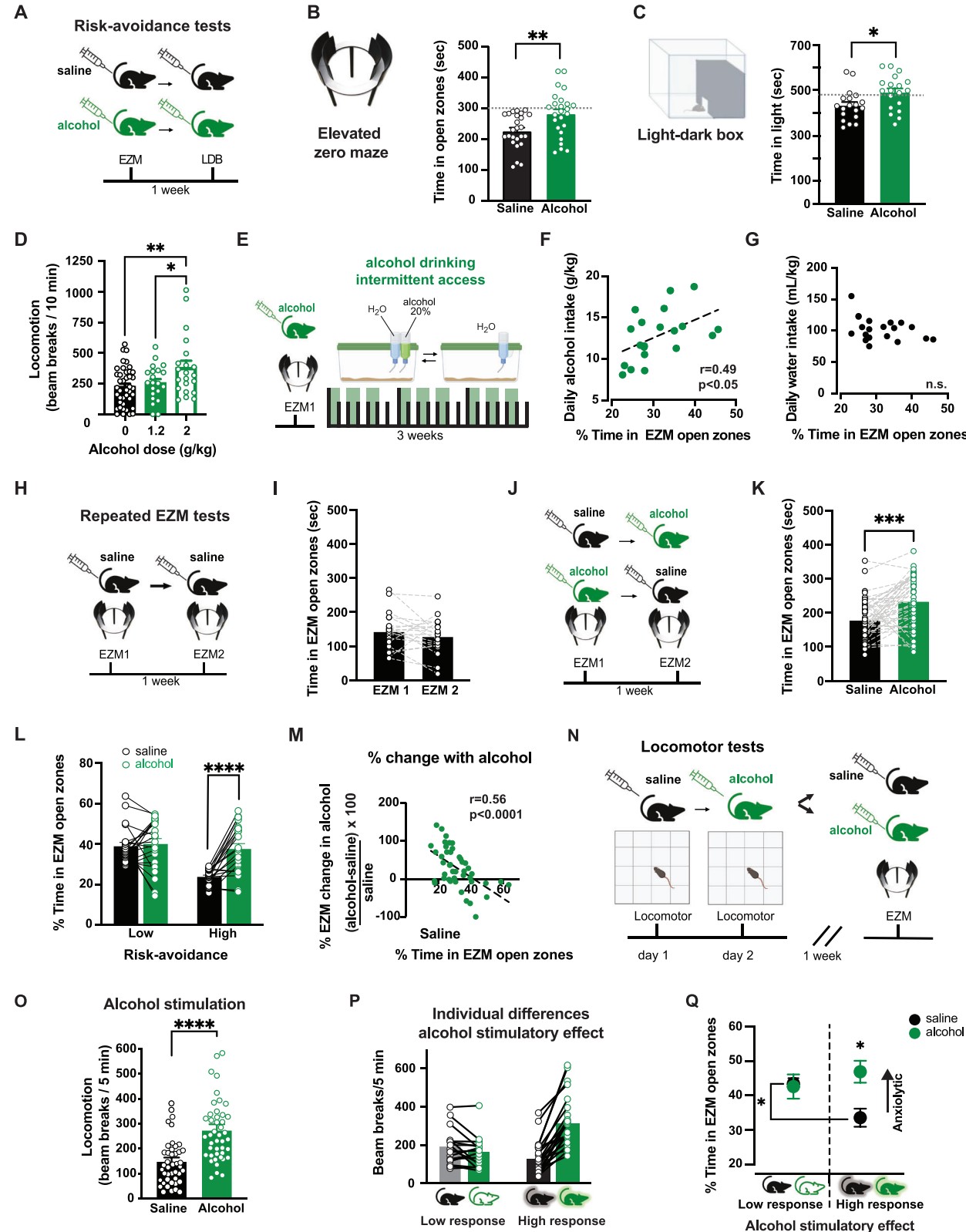

the time spent in open zones for mice with high risk-avoidance (those exhibiting less time in the open zones after saline administration) and had no effect in those with low risk-avoidance (Fig. 1L). This differential effect of alcohol was observed when mice fell into "high" or "low" groups based on the median (172 s; Fig. 1L) or the mean EZM performance after saline (184 s, Supplemental Fig. 2F, G), reinforcing the observation that alcohol's anxiolytic effects are stronger in mice with a high risk-avoidance phenotype. Sorting these data by sex yields similar findings (Supplemental Fig. 2H).

Furthermore, the repeated EZM tests allowed for calculation of the percent change in EZM performance following alcohol in each mouse (time in open zones following alcohol - time in open zones following saline, normalized by time in open zones following saline). The percent change in EZM performance between saline and alcohol

**Fig. 1 | Individual variation in the expression of anxiety-like behaviors and relief by alcohol in C57BL/6J mice. A** Experimental design for C57BL/6J mice. **B**, **C** Time spent in EZM open zones (two-tailed t-test t(48) = 3.0, $p = 0.004$, $n = 25$ mice/group) and LDB (t(36) = 2.5, $p = 0.017$, $n = 19$ mice/group) for mice receiving saline (black) or 1.2 g/kg alcohol (green). Symbols represent individual mice data; bars and lines represent mean ± SEM of each group. **D** Locomotor assessment after saline, 1.2, or 2 g/kg alcohol. Only 2 g/kg alcohol significant increased beam breaks compared to saline ($p = 0.007$) and to 1.2 g/kg alcohol ($p > 0.05$; 1-Way ANOVA F(2,80) = 5.27, $n = 40$ saline mice, 21 mice with 1.2 g/kg and 21 mice with 2 g/kg). Symbols represent individual mice data; bars and line represent mean ± SEM of each group. **E** Mice were tested on EZM after alcohol (1.2 g/kg) followed by 2-bottle intermittent alcohol drinking task. **F** Positive correlation between EZM performance after alcohol and voluntary alcohol intake (linear regression fit F(1,17) = 5.82, $p = 0.027$, $n = 19$ mice). **G** No correlation was found between EZM performance and water intake (linear regression fit F(1,17) = 2.73, $p > 0.05$, $n = 19$ mice). **H** Mice were tested on EZM twice following saline, one week apart. **I** EZM performance was similar across saline tests (two-tailed t-test t(20) = 1.31, $p = 0.21$, $n = 21$ mice). **J**, **K** Separate cohort tested twice on EZM following saline (black) or alcohol (1.2 g/kg; green), using within-subjects counterbalanced design. Time in EZM open zones was higher after alcohol than saline (two-tailed t-test t(42) = 4.23, $p = 0.0001$, $n = 43$). **L** Median split of time spent in EZM open zones after saline (173 s) was used to sort mice into "high" and "low" risk-avoidance groups. Only "high" group shows increased time in EZM open zone following alcohol compared to saline (RM 2-way ANOVA Group x Alcohol interaction: F(1,41) = 11.13, $p = 0.002$; main effect of Alcohol and Group:

Fs(1,41) < 13.22 p = 0.0003, post hoc test $p < 0.0008$; $n = 43$). Symbols represent individual mice data; bars and lines represent ± SEM of each group. **M** Inverse correlation between time spent in EZM open zones after saline and percent change in time after alcohol (linear regression fit F(1,41) = 18.74, $p < 0.0001$, $n = 43$). Symbols represent individual mice data. **N** Mice were tested in locomotor boxes after saline and 2 g/kg alcohol and then tested on EZM following saline OR alcohol (1.2 g/kg). **O** Beam-breaks in locomotor boxes were higher following alcohol than saline (two-tailed t-test t(43) = 5.24, $p < 0.0001$, $n = 44$ mice). Symbols represent individual mice data; bars and lines represent mean ± SEM of each group. **P** Low response mice show no increase after alcohol and high response mice show increase (RM 2-way ANOVA Drug x Group interaction: F(1,42) = 25.58, $p < 0.0001$; post-hoc test Saline vs Alcohol locomotion $p < 0.0001$ in "high" group, $n = 28$ mice and $p > 0.05$ in "low" group; $n = 16$ mice). Symbols represent individual mice data; bars and lines represent mean ± SEM of each group. **Q** Low response mice showed similar EZM performance after saline and 1.2 g/kg alcohol. High response mice showed less time in EZM open zones after saline than low response and time in open is increased after 1.2 g/kg alcohol compared to saline (RM 2-way ANOVA group x EZM interaction: F(1,42) = 4.52, $p = 0.039$; the difference in saline EZM between low vs high t(20) = 2.10, $p = 0.049$, $n = 22$ mice; difference in saline vs alcohol in high response mice t(26) = 3.19, $p = 0.004$, $n = 28$ mice). Symbols and line represent mean ± SEM of each group. For all panels, *$p < 0.05$, **$p < 0.01$, ****$p < 0.0001$. Cartoons in panels (**B**, **C**, **N**) are modified BioRender template license Alvarez, V. (2024) BioRender.com/m14d797.

tests for each mouse revealed a negative correlation with the time spent in the open zones after saline (i.e., baseline risk avoidance; Fig. 1M). This independent analysis, which didn't rely on post-hoc categorization of mice, confirmed that mice which spend less time in the open zones following saline (high risk-avoidance) exhibited a larger change following alcohol administration. As control, we performed these same analyzes in the data from mice that received saline twice (Fig. 1I) and did not see any differences in performance between the first and second test (Supplemental Fig. 2I, J). Collectively these results emphasize the selective influence that alcohol has on the subgroup of mice characterized by heightened risk-avoidance behavior.

## Correlation between the stimulatory and anxiolytic effects of alcohol

For a second experiment addressing the relationship between risk-avoidance behavior and the potency of alcohol relief, we investigated whether the group of mice that are more sensitive to alcohol's stimulatory effects are distinct or overlapping with the group of mice with high risk-avoidance and high alcohol sensitivity.

Alcohol's stimulatory effects were assessed in locomotor chambers. On two consecutive test days, mice received either saline or alcohol (2 g/kg, i.p.), in a randomized order. This dose of alcohol was chosen because it produces maximal alcohol stimulation in this mouse strain[30] (Fig. 1D and Supplemental Figure 7D). One week later, mice were subjected to a single EZM test following either saline or alcohol (1.2 g/kg; Fig. 1N). Overall, mean locomotion was higher in mice following alcohol compared to saline administration (Fig. 1O). Consistent with prior reports, some mice displayed a robust increase in locomotion after alcohol (which we classified as high-responders), while others exhibited minimal or no increase from saline locomotor levels (which we classified as low-responders; Fig. 1P). When tested in the EZM, high-response mice that received alcohol spent more time in open zones compared to those that received saline. To the contrary, low-response mice showed similar times in open zones after alcohol and saline (Fig. 1Q). These observations suggest a positive association between alcohol stimulation and anxiolysis.

When comparing mice that received saline, high-response mice also exhibited less time in open zones than low-response mice. This finding confirms and expands on the results of the repeated EZM experiment and indicates that mice with a risk-

avoidance phenotype show stronger anxiolytic and stimulant alcohol responses.

There was a negative correlation between alcohol stimulation and time in the open zones of the EZM following saline and a positive correlation between alcohol stimulation and time in the open zones following alcohol (Supplemental Fig. 3A–D). This directional shift in correlation results from mice with high stimulation undergoing the most significant change in EZM open zone exploration when administered alcohol, transitioning from high risk-avoidance after saline to low risk-avoidance after alcohol. No correlations were seen within mice between locomotion following saline versus alcohol indicating that the observed phenomenon was not simply hyperactivity (Supplemental Fig. 3E). Nor was there a correlation between locomotion following saline and time spent in the open zones of the EZM following saline (Supplemental Fig. 3F).

We explored whether the association between stimulant and anxiolytic effects was generalizable. We tested whether cocaine, another stimulant drug, could increase time exploring the EZM open zones. We found that while cocaine (10 mg/kg) increased velocity, it did not increase time spent in the EZM open zones (Supplemental Fig. 3G), demonstrating a dissociation between locomotor stimulation and decreased risk avoidance[41]. Thus, increased time in EZM open zones is not necessarily a consequence of heightened locomotion, underscoring the specificity of the observed correlation with alcohol's effects.

Collectively, these findings present strong evidence supporting an association between the anxiolytic and stimulant effects of alcohol, hinting at potential shared underlying mechanisms. The function of dopamine D1 receptors (D1R) was shown to be critical for mediating alcohol-induced stimulation in our previous study[35], leading us to examine the role of the D1R in the anxiolytic effects of alcohol.

## D1 activation in the striatum is required for alcohol anxiolytic effects

Pharmacological experiments were conducted to examine the role of D1Rs in mediating alcohol's anxiolytic effects. Systemic pre-treatment with a low dose of the D1-like receptor antagonist SCH-23390 (0.03 mg/kg, i.p.) effectively blocked the alcohol-induced increase in time spent in the open zones of the EZM, while leaving overall mouse movement unaffected (Fig. 2A, Supplemental Fig. 4). These findings suggest the involvement of D1Rs in mediating alcohol's anxiolytic

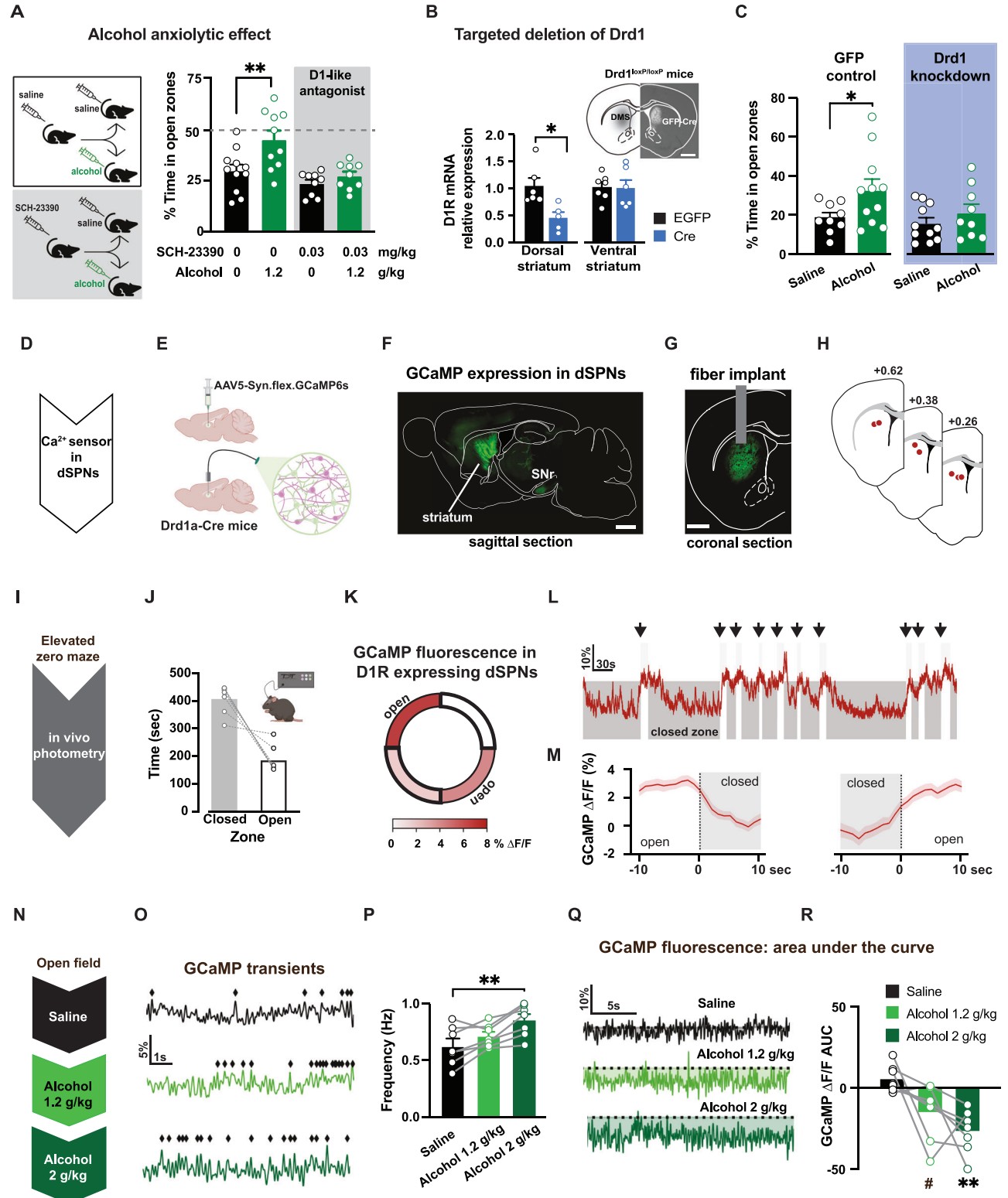

effects. However, the systemic administration of the antagonist limits the ability to pinpoint the specific brain region(s) involved.

We employed a genetic approach in the dorsomedial striatum to test the regional specificity of the involvement of D1Rs. Viral vectors expressing Cre recombinase were injected in mice with conditional Drd1a alleles to selectively delete the Drd1a gene in the dorsomedial striatum. Likely due to partial coverage of the viral vector expression, this targeted genetic manipulation resulted in a 55 ± 10% reduction in Drd1a mRNA in the dorsal striatum, with no significant change observed

in the ventral striatum (Fig. 2B). When tested on the EZM, mice with Drd1a knockdown in the dorsomedial striatum exhibited a diminished anxiolytic response to alcohol and showed no significant increase in time spent in the open zones compared to saline (Fig. 2C). Control mice that received a GFP-expressing vector in the dorsomedial striatum showed an increase in the time spent in the open zones after alcohol, confirming alcohol anxiolytic effects are intact in this transgenic line. This genetic strategy revealed the contribution of D1 receptors in the dorsomedial striatum to mediating alcohol's anxiolytic effects.

**Fig. 2 | Open zone exploration engages striatal projection neurons expressing D1R, which are required for alcohol's anxiolytic effect. A** Mice were pre-treated with saline or dopamine D1-like antagonist SCH-23390 (0.03 mg/kg) and tested on the EZM after alcohol (1.2 g/kg) or saline (2-way ANOVA pretreatment x EZM drug interaction: $F(1,37) = 17.06$, $p = 0.0002$, $n = 39$ mice). Only saline pre-treated mice showed differences in EZM performance between saline and alcohol (post hoc Sidak test $p < 0.01$, $n = 10,11$ mice/group). SCH-23390 pre-treated mice showed similar EZM performance (post hoc Sidak test $p > 0.05$; $n = 9$ mice/group). **B** Drd1 mRNA levels in dorsal and ventral striatum from Drd1$^{loxP/loxP}$ mice with bilateral intracranial injections of either EGFP- or Cre-expressing vectors in dorsomedial striatum. Lower Drd1 mRNA levels in dorsal, but not ventral, striatum of mice with Cre-vector compared to EGFP (dorsal: $t(9) = 3.10$, $p = 0.013$, ventral: $t(9) = 0.26$, $p = 0.801$, $n = 6$ mice/group, t-test). Inset, Fluorescent image of coronal section showing Cre expression in dorsal striatum of Drd1$^{loxP/loxP}$ mice. Scale bar is 1 mm. **C** Mice expressing Cre in dorsal striatum displayed similar EZM performance after alcohol (1.2 g/kg) compared to saline (2-way ANOVA $F(3,41) = 3.60$, $p = 0.021$; post hoc Sidak test saline vs. alcohol: $p = 0.03$ control EGFP mice and $p > 0.05$ for Cre-expressing mice; $n = 9,11,11,9$ mice/group). **D, E** Ca$^{2+}$ sensor GCaMP6s expressed in D1R-containing neurons in dorsal striatum of Drd1a-Cre mice. Cartoon made from modified BioRender template (license Alvarez, V. 2024 BioRender.com/f17i932) **F, G** Fluorescent images of sagittal and coronal sections showing on left GCaMP6s expression in dorsal striatum and axon projections in substantia nigra (SNr) and on right, the placement of photometry fiber relative to GCaMP expression. Similar images were repeatedly and independently obtained from the six mice used in this experiment. Scale bar is 1 mm. **H** Fiber placement locations obtained from post-mortem histology. **I** Fiber photometry measurements of GCaMP6s signals were made while mice were tested in the EZM. **J** Mice with fiber implants spent less time in the open than closed zones ($n = 6$ mice). **K–M** $\Delta$F/F signals were higher during the exploration of open zones compared to closed ($n = 6$ mice). **N** GCaMP6s $\Delta$F/F signals were measured after mice received increasing doses of alcohol (0, 1.2 and 2 g/kg). **O, P** Increased frequency of transients were observed as alcohol concentration increased (RM 1-way ANOVA $F(2,12) = 8.62$, $p < 0.01$; post hoc Tukey test saline vs. 2 g/kg alcohol, $p = 0.005$, $n = 6$). **Q, R** Overall GCaMP fluorescence dropped after alcohol (RM 1-way ANOVA main effect of Dose: $F(2,12) = 7.85$, $p = 0.007$; post hoc Tukey test saline vs 2 g/kg alcohol $p = 0.007$; saline vs 1.2 g/kg alcohol $p = 0.06$, $n = 6$ mice). Data presented as mean ± SEM, $^{\#}p = 0.06$, $^*p < 0.05$, $^{**}p < 0.01$.

## Recruitment of D1R-expressing striatal neurons during open zone exploration

The activity of D1R expressing neurons within the dorsomedial striatum was assessed during the EZM task using in vivo fiber photometry measurements. For these experiments, the fluorescent calcium sensor GCaMP6s was expressed using a Cre-dependent viral vector in Drd1a-Cre mice to target D1 expressing spiny projection neurons (Fig. 2D–G). During behavioral experiments, being tethered to the fiber did not disrupt EZM performance in an observable way and mice spent comparable time exploring the open zones (Fig. 2I, J).

We found that the overall GCaMP6s signal ($\Delta$F/F) was higher during the exploration of the open zones compared to the closed zones (Fig. 2K). The photometry signal showed evident changes at the times when mice transitioned between closed and open zones (Fig. 2L). When the GCaMP6s signals were aligned to the transition between zones, the average $\Delta$F/F increased as the mouse moved from closed to open zones and decreased as the mouse moved from open to closed zones (Fig. 2M). These experiments indicate that D1R expressing SPNs, as a population, show increased calcium events when mice enter the anxiogenic zone of a maze. This suggests exploration of the risk zones might be promoted by activation of D1 SPNs.

## Alcohol acutely promotes calcium transients in D1R- expressing striatal neurons

Again using in vivo fiber photometry of GCaMP6s in dorsal striatum, we assessed population calcium events in D1R expressing striatal neurons following administration of alcohol at the two doses used previously in this study. Alcohol at 1.2 g/kg showed a modest increase in GCaMP6s activity compared to saline administration, and 2 g/kg alcohol produced a significant increase in the frequency of population calcium events in D1R expressing neurons in the dorsomedial striatum (Fig. 2N–P). Overall, alcohol administration enhanced the frequency of calcium transients in D1R-expressing neurons in a dose-dependent manner.

We noted that in addition to increased frequency of spontaneous transients, alcohol produced a slow decrease in the signal baseline, quantified in the area-under-the-curve (AUC) measurements (Fig. 2Q, R). This change in baseline fluorescence is likely to reflect hemodynamic changes, as recently reported[42], possibly in response to neuronal activation and/or alcohol- induced vasodilation.

Collectively, these studies reveal three interconnected findings: a) D1 SPNs are activated during animal exploration of the risk zones of the EZM; b) acute alcohol exposure promotes activation of D1 SPN; and c) the anxiolytic effects of alcohol depend on the activation of D1Rs specifically in the dorsal striatum. Altogether, these findings propose a potential mechanism through which alcohol induces its anxiolytic effects.

## Risk-avoidance is associated with high ratio of striatal D1 to D2 dopamine receptors

We hypothesized that the individual variability in the EZM performance and alcohol relief could be due to differences in gene expression of D1R across the striatum. To test this, alcohol-naive mice were tested on the EZM after saline administration and striatal tissue samples were collected for assessment of gene expression levels for the two most highly expressed dopamine receptors in the striatum, D1R and D2R (Fig. 3A). Neither *Drd1* nor *Drd2* mRNA levels in the dorsomedial striatum were correlated with time in open zones during EZM. However, calculating the ratio *Drd1* over *Drd2* mRNA for each mouse revealed a negative correlation with time spent in the risk zones of the EZM (Fig. 3A). Sorting these data by sex yields negative correlations for both female and male datasets (Supplemental Fig. 5A, B). When mice were administered alcohol before the EZM, and tissue was later collected, again neither *Drd1* nor *Drd2* mRNA levels in the dorsomedial striatum were correlated with time in open zones, but the ratio of receptor mRNA was positively correlated with time spent in the risk zones (Supplemental Fig. 5A–D).

Interestingly, the alcohol potency on striatal tissue was also correlated to EZM performance after saline. Alcohol-naive male mice with a risk-averse phenotype (less time spent on EZM open zones) showed stronger in vitro effect of alcohol on evoked dopamine release in the dorsal striatum, compared to mice with more time spent on EZM open zones (Fig. 3B). Using fast-scan cyclic voltammetry in brain slices, we found that the magnitude of the acute alcohol effect was variable (Supplemental Fig. 5H) and correlated with EZM performance in male mice. Males that spent less time in the open zones showed a stronger in vitro alcohol effect (Fig. 3B). No correlation was observed in tissue from female mice, alcohol (80 mM) had a smaller overall effect on evoked dopamine signals in the striatal slices from female mice compared to those from male mice (Fig. 3B; Supplemental Fig. 5G). This is one of the few analyzes in the current study that revealed a sex difference. In agreement with recent findings in the amygdala[43], these findings suggest that female brain circuitry is less sensitive to alcohol, and could explain the higher levels of alcohol consumption observed in female compared to male mice.

Together, these in vitro observations argue in support of molecular and biological mechanisms underlying the individual variability in the effect of alcohol.

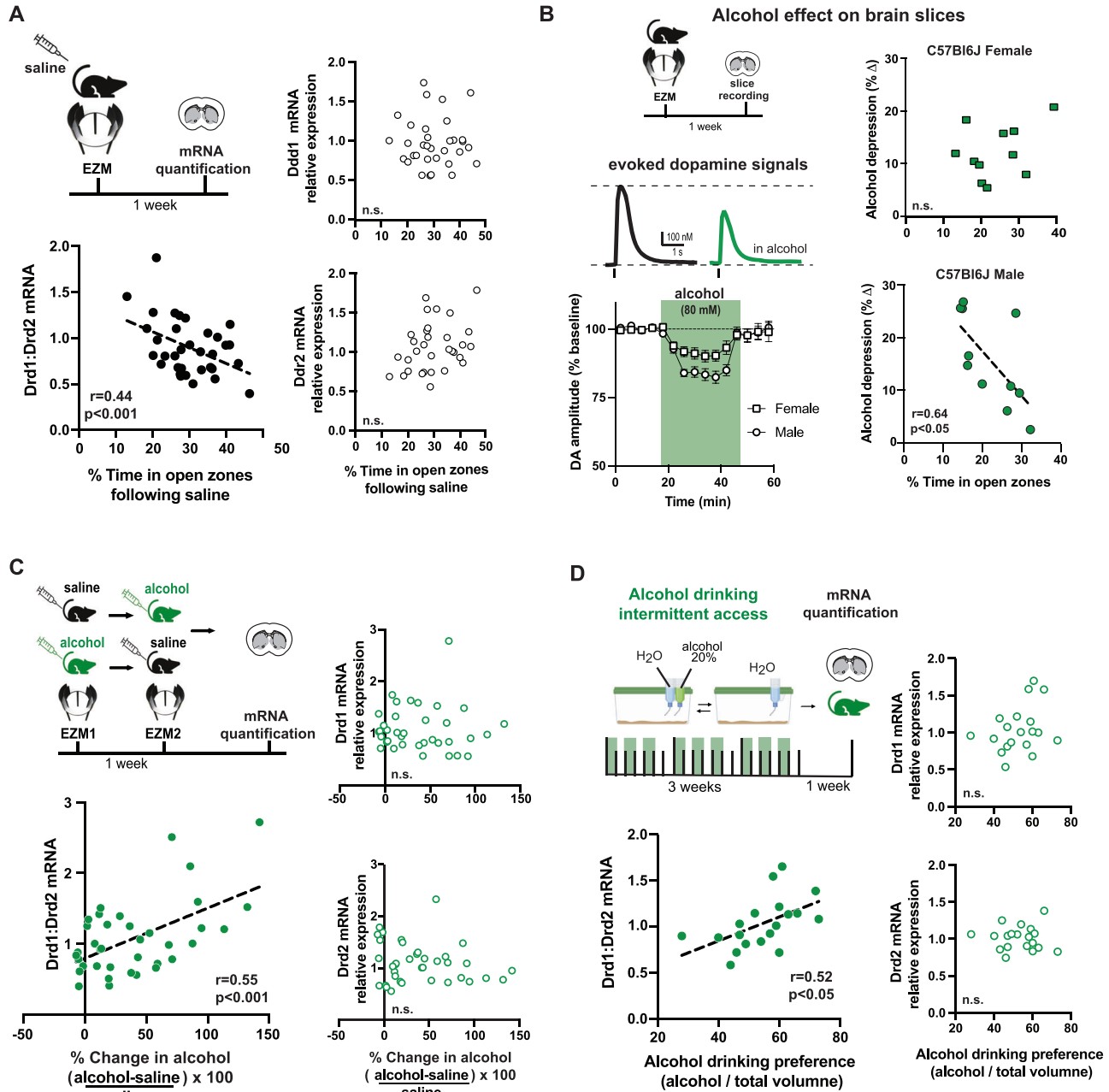

**Fig. 3 | High ratio of striatal Drd1 over Drd2 mRNA expression is associated with risk-avoidance, stronger alcohol relief and high alcohol preference. A** Mice were tested on EZM after saline and striatal samples collected for mRNA quantification. Neither Drd1 nor Drd2 mRNA levels in the striatum were correlated with EZM performance (linear regression fit Drd1: F(1,32) = 1.06, p = 0.311 and Drd2: F(1,32) = 2.47, p = 0.1259). The ratio of Drd1/Drd2 mRNA showed negative correlation with time in EZM open zones (F(1,32) = 7.59, p < 0.00096, n = 34). **B** Mice were tested on EZM and striatal brain slices were prepared for in vitro analysis of alcohol's effect on evoked dopamine. Sample traces of evoked dopamine signals at baseline (black) and after application of 80 mM alcohol (green). Time course of the change in amplitude of evoked dopamine in slices from male (circle) and female (square) mice. Symbols and lines represent mean ± SEM of evoked dopamine amplitude normalized to baseline before alcohol application. Male mice showed larger depression of evoked dopamine than females (RM 2-way ANOVA significant interaction Alcohol x Sex F(14,420) = 3.51 p < 0.0001, n = 14,18 slices; 11 mice/sex). While there was no relationship between EZM performance and alcohol's in vitro

effect on dopamine in females, an inverse relationship was found in males (linear regression fit male: F(1,9) = 7.51, p = 0.023, n = 11 males; female: F(1,10) = 2.1, p = 0.178, n = 12 females). **C** Following repeated EZM experiments (Fig. 1I), striatal tissue samples were collected for assessment of Drd1 and Drd2 mRNA levels. While neither Drd1 nor Drd2 mRNA levels were correlated with EZM performance following alcohol (linear regression fit Drd1: F(1,35) = 2.19, p = 0.148; Drd2: F(1,35) = 0.97, p = 0.331), the ratio of Drd1/Drd2 was correlated with EZM performance following alcohol (F(1,35) = 15.10, p < 0.0004, n = 36 mice). **D** Mice were given access to 20% alcohol for 3 weeks on two-bottle choice task, and dorsal striatal tissue collected for mRNA quantification. While neither Drd1 levels nor Drd2 mRNA levels were correlated with alcohol drinking preference (linear regression fits Drd1: F(1,17) = 1.94, p = 0.182; Drd2: F(1,17) = 0.02, p = 0.889), the ratio of Drd1/Drd2 was correlated with alcohol preference (F(1,17) = 6.29, p = 0.023, n = 19 mice). EZM and cage cartoons in panels (**A**, **B**, **C**, **D**) are modified BioRender template license Alvarez, V. (2024) BioRender.com/m14d797.

## Higher ratio of striatal D1 to D2 receptors in mice with more alcohol relief and drinking preference

Mice underwent repeated EZM a week apart following saline or alcohol (1.2 g/kg) injections, in a randomized order. One week later striatal tissue samples were collected for assessment of dopamine D1R and D2R mRNA (Fig. 3C, top left). Again, there was no correlation between the EZM performance and the expression of Drd1 mRNA nor Drd2 mRNA in the dorsal striatum. Interestingly, a positive correlation was found when examining the ratio of Drd1 to Drd2 mRNA levels. Mice with a larger percent change in time in the open zones of the EZM following alcohol showed higher ratio of striatal Drd1 to Drd2 mRNA expression (Fig. 3C, bottom left). Sorting these data by sex yields positive correlations for both female and male mice between the ratio of dopamine receptor expression and the percent change by alcohol (Supplemental Fig. 5I, J).

Similar experiments were performed assessing the degree of alcohol preference using an intermittent access two-bottle choice paradigm for 3 weeks (9 sessions of 24-h access to a 20% alcohol solution and water; food was available *ad libitum*). Despite the lack of a correlation between alcohol preference and the individual levels of Drd1 or Drd2 mRNA in the dorsal striatum, the ratio of Drd1 to Drd2 mRNA expression was positively correlated with alcohol drinking preference (Fig. 3D).

## Manipulation of striatal D1 and D2 receptor ratio in mice

To test the causality of the D1/D2 receptor ratio in the behavioral expression of risk avoidance and the alcohol response, we turned to a genetic approach in which we lowered the D2 receptors in the striatum, changing the ratio of D1/D2 receptors. We predicted that increasing the ratio of D1 to D2 receptors in the striatum would promote a baseline risk avoidance phenotype in mice, and in turn, enhance alcohol's anxiolytic effects and drinking.

A single allele deletion of *Drd2* gene was targeted to striatal projection neurons by crossing Drd2$^{LoxP/LoxP}$ with Adora2a-Cre mice to produce iSPN-Drd2HET and littermate controls (Fig. 4A). *Drd2* mRNA levels were reduced by half in both dorsal and ventral striatum samples from iSPN-Drd2HET mice (50 ± 7% and 49 ± 3%, respectively; Fig. 4B). The levels of Drd1 mRNA remained unchanged (Fig. 4B), resulting in a doubling of the ratio of Drd1 to Drd2 mRNA expression throughout the striatum of iSPN-Drd2HET mice (Fig. 4C). The dopamine receptor availability was assessed using radioligand binding for D2-like and D1-like receptor families (Fig. 4D, E, respectively). Binding by the D2-like receptor ligand [3H]raclopride was reduced by 45 ± 3% in iSPN-Drd2HET mice compared to littermate controls. Binding for the D1-like receptor ligand [3H]SCH-23390 was slightly increased in iSPN-Drd2HET striatum and consequently, the ratio of D1 to D2 receptor ligand binding was double that of controls across all striatal subregions of the iSPN-Drd2HET mice (Fig. 4F).

## Mice with high D1 to D2 receptor binding display risk-avoidance phenotype

iSPN-Drd2HET mice of both sexes were tested in a battery of behavioral tests and compared to littermate controls (Drd2$^{flox/wt}$ and Drd2$^{flox/flox}$). Before implementing tasks that rely on explore-avoid conflict, we assessed the baseline locomotor between the genotypes and found no difference in locomotor activity when measured in the home cage (Fig. 4G, H). However, when tested in a novel arena, iSPN-Drd2HET mice showed reduced distance traveled, indicating a selective reduction in exploratory behavior, without locomotor impairment (Fig. 4I, Supplementary Fig. 6A, B). iSPN-Drd2HET mice also spent less time exploring the open zones of the EZM and the light compartment of the light-dark box, than littermate controls (Fig. 4J, K). Lastly, fasted iSPN-Drd2HET mice also had longer latencies to feed in a novel environment (Fig. 4L). The overall composite score calculated across these measurements showed higher risk-

avoidance phenotype in iSPN-Drd2HET compared to littermate controls (Fig. 4M).

We also tested mice with double allele deletion of *Drd2* gene in D2R-expressing SPNs (iSPN-Drd2KO) that are expected to have an even larger D1/D2 ratio in the striatum than the iSPN-Drd2HET mice. We previously showed that these mice have risk-averse tendencies[44] but also motor impairments[45], which could be a confounding factor in the exploration tasks. In fact, iSPN-Drd2KO mice exhibited a more pronounced phenotype than heterozygotes in novel open field exploration (Supplemental Figure 6A). Interestingly, iSPN-Drd2KO mice showed similar risk avoidance behavior (Supplementary Fig. 6B–D) and composite scores as the iSPN-Drd2HET mice (Supplemental Figure 6E). Thus, functional loss of a single Drd2 allele is sufficient for full expression of the risk-avoidance phenotype.

Serum corticosterone levels were elevated in both iSPN-Drd2HET and KO mice compared to littermate controls, indicative of a stress response (Fig. 4N, Supplemental Figure 6F). We found no differences in sucrose preference measured with a 1% solution, arguing against an anhedonia phenotype (Fig. 4O, Supplemental Figure 6G). Taken together, these data show that a partial reduction of striatal D2R levels, that doubles the ratio of D1/D2 receptor surface expression, was sufficient to promote a high risk-avoidance phenotype in mice.

## Enhanced anxiolytic potency of alcohol in mice with high D1 to D2 receptor ratio

iSPN-Drd2HET mice, which have a high ratio of D1/D2 receptors, showed larger increase in open zone exploration following alcohol compared to controls (64 ± 11% vs 16 ± 7% in control; Fig. 5B). This effect was due to the low saline EZM performance of iSPN-Drd2HET mice and the absolute time spent in the open zones after alcohol did not differ between genotypes (control: 47 ± 3% vs. iSPN-Drd2HET: 43 ± 5% and iSPN-Drd2KO: 47 ± 3%; Fig. 5A), indicating a possible ceiling effect and/or that alcohol acts to erase the phenotypic difference in risk-aversion behavior among mice. Homozygous iSPN-Drd2KO also showed a similar enhancement in the alcohol response (Supplemental Figure 7B).

Despite the differences in the behavioral response to alcohol, blood ethanol concentration was similar across genotypes following administration of the same alcohol dose used in the EZM (1.2 g/kg; Fig. 5C, Supplemental Figure 7C).

Systemic administration of a D1-like receptor antagonist blocked the ability of alcohol to increase time in open zones of the EZM in iSPN-Drd2HET mice (Fig. 5D, E), similar to the findings in control mice reported in Fig. 2A. This result provides evidence that the molecular mechanisms underlying alcohol-induced anxiolytic properties in mice with a manipulated ratio of D1 to D2 receptors are similarly dependent on D1R activation as we found with wildtype mice. Altogether, these findings indicated that mice with high striatal D1/D2 receptor ratio have enhanced sensitivity to alcohol's anxiolytic effects.

The stimulant effects of alcohol were also enhanced in iSPN-Drd2HET mice with high D1/D2 receptor ratio, in agreement with our previous findings with the double allele deletion mice[35]. There was an upward shift in the dose response, which peaked at 2 g/kg (Fig. 5F, Supplemental Figure 7D). Note that 1.2 g/kg alcohol, the dose used in EZM, does not produce significant locomotor stimulation in iSPN-Drd2HET, again ruling out a possible confound between the stimulant effects and the EZM performance. Here we found an association between alcohol stimulation, its anxiolytic potency, and the baseline risk-aversion phenotype. Mice with the largest alcohol stimulation showed the highest risk-aversion at baseline and the largest change in the EZM exploration after alcohol (Fig. 5G).

## Propensity to punishment-resistant alcohol drinking in mice with high ratio of D1 to D2 receptors

Operant alcohol drinking was next assessed during social housing using the Intellicage System. iSPN-Drd2HET and littermate control

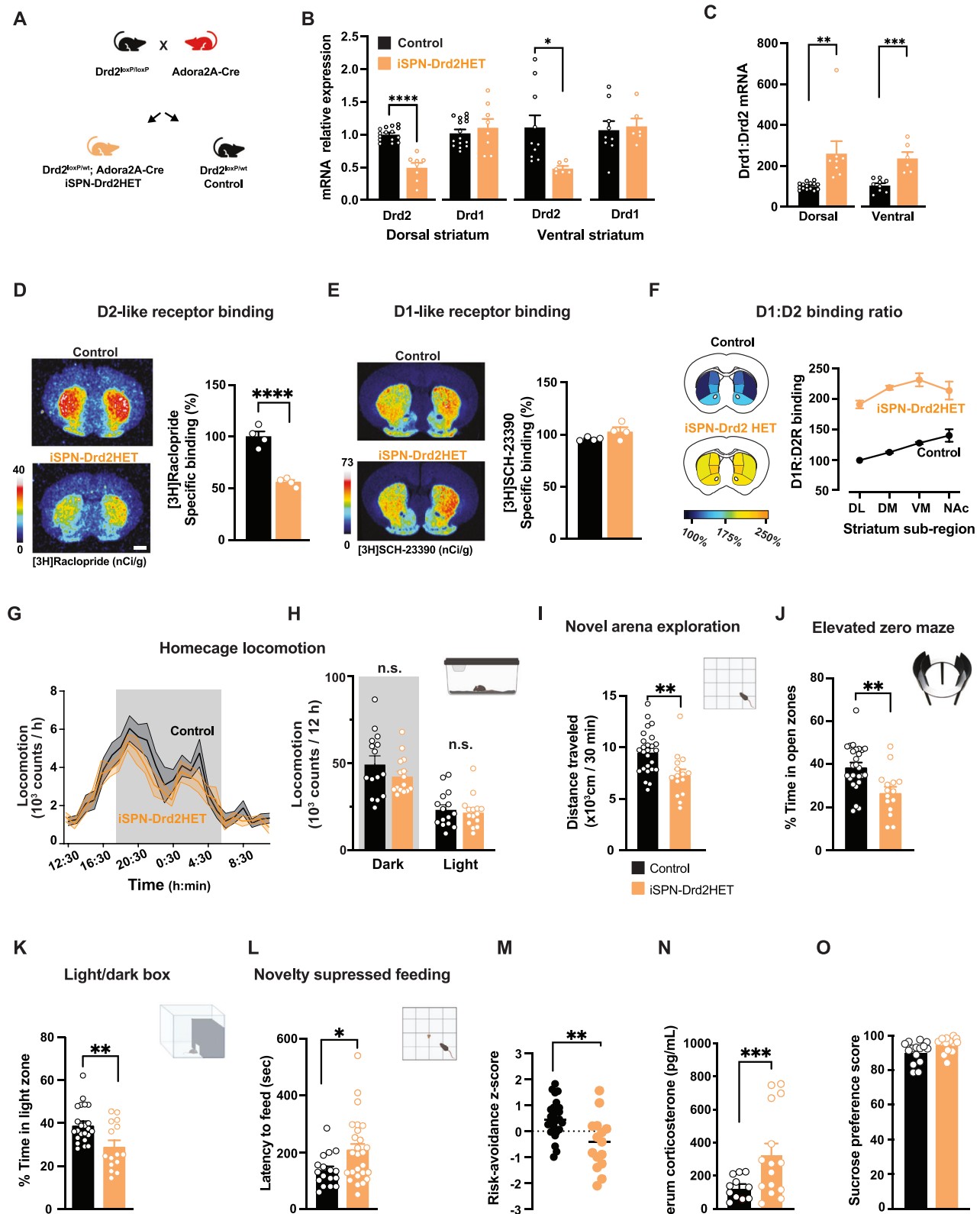

mice were housed together in groups of 16–18. Mice were given constant access to water and intermittent (24 h, every other day) access to alcohol (20%). Each social-housing cage is equipped with four operant test chambers (2 for water and 2 for 20% alcohol) and entry into each operant chamber is measured via unique RFID tags. Mice were trained to perform a fixed ratio of 3 nose-pokes to gain access to sipper tubes. Licks and nose-pokes were measured. Mice from both genotypes

displayed circadian rhythm in water and alcohol consumption, with higher consumption during the light-off phase of the light cycle (Supplemental Figure 7E). The mean number of licks per session for alcohol was similar for both genotypes (Fig. 5H) and the range of individual variability was similar as well (Supplemental Figure 7F). On water only days, the mean number of licks per session for water was similar for both genotypes (Fig. 5I). There was also no difference in

**Fig. 4 | Generation of mice with high ratio of striatal D1 over D2 receptor availability promoted expression of risk-avoidance phenotype. A** Mice with low levels of striatal dopamine D2R (iSPN-Drd2HET) and littermate controls were generated by breeding Drd2[loxP/loxP] with Adora-2A-Cre mice. **B** iSPN-Drd2HET showed decreased Drd2 mRNA in dorsal and ventral striatum, but no significant change in Drd1 mRNA (2 W ANOVA dorsal striatum: receptor x genotype interaction $F_{(1,21)} = 39.36$, $p < 0.0001$, main effect of receptor type $F_{(1,21)} = 43.34$, $p < 0.0001$ and genotype $F_{(1,21)} = 6.62$, $p = 0.018$, $n = 8,15$ mice/group; ventral striatum: receptor x genotype interaction $F_{(1,13)} = 7.47$, $p = 0.017$, main effect of receptor type $F_{(1,13)} = 7.31$, $p = 0.018$, $n = 6,9,10$ mice/group). **C** Drd1/Drd2 mRNA ratio is higher in iSPN-Drd2HET (orange) compared to littermate controls (black), both in dorsal and ventral striatum (main effect of genotype (2-way ANOVA main effect of genotype $F_{(1,34)} = 26.49$, $p < 0.0001$; dorsal: $t_{(21)} = 3.73$, $p = 0.001$, $n = 8-15$ mice/genotype and ventral: $t_{(13)} = 4.97$, $p = 0.0003$, $n = 6-10$ mice/genotype). **D, E** Representative saturation plots of D2-like ligand [3H]raclopride (**D**) and D1-like ligand [3H]SCH-23391 **E** binding to coronal brain sections from control (black) and iSPN-Drd2HET mice (orange; scale bar is 1 mm). Bar graphs show specific binding quantification across replicates (t-test control vs. iSPN-Drd2HET: $t_{(6)} = 9.73$, $p < 0.0001$ for raclopride; $t_{(6)} = 1.90$, $p = 0.106$ for SCH-23390; $n = 4$/genotype). **F** Ratio of ligand binding for D1-like and D2-like ligands in iSPN-Drd2HET and littermate control mice ($n = 4$ mice/genotype). **G** Locomotion in the home cage is similar for iSPN-Drd2HET (orange) and littermate control mice (black) over 24-hours (RM 2-way ANOVA $F_{(23,644)} = 0.73$, $p > 0.05$). Data collapsed cross "light" and "dark" phase of light-cycle are also similar (**H**) RM 2-way ANOVA $F_{(1,28)} = 1.76$, $p > 0.05$, $n = 15$/genotype). **I** iSPN-Drd2HET mice showed reduced exploration of a novel arena relative to controls (t-test $t_{(39)} = 3.27$, $p = 0.002$, $n = 15,26$ mice/genotype). **J** iSPN-Drd2HET mice show decreased time in EZM open zones (t-test $t_{(39)} = 3.48$, $p = 0.001$, $n = 15,26$ mice/genotype), **K** decreased time spent in light compartment of light-dark box (t-test $t_{(36)} = 3.14$, $p = 0.003$, $n = 15,23$ mice/genotype), and **L** longer latency to feed on the novelty suppressed feeding task in iSPN-Drd2HET (t-test $t_{(41)} = 2.35$, $p = 0.024$, $n = 17,26$ mice/genotype). **M** Composite z-scores across all tasks for iSPN-Drd2HET (orange) and littermate control (black) (t-test $t_{(39)} = 3.47$, $p = 0.001$, $n = 15,26$ mice/genotype). **N** Serum corticosterone levels are higher in iSPN-Drd2HET mice (orange) compared to littermate controls (black) (t-test $t_{(24)} = 3.88$, $p = 0.0007$, $n = 11,15$ mice/genotype). **O** No genotypic difference in a sucrose anhedonia test (t-test $t_{(27)} = 1.92$, $p > 0.05$, $n = 12,15$ mice/genotype). Data presented as mean ± SEM, *$p < 0.05$ **$p < 0.01$, ***$p < 0.001$, ****$p < 0.0001$. Cartoons in panels h, i, j, k, l are modified BioRender template license Alvarez, V. (2024) BioRender.com/m14d797.

water intake between genotypes on the days that water and alcohol were concurrently available (Supplemental Figure 7G).

After three weeks of testing, we assessed the mouse response to the adulteration of alcohol with a bitter tastant. Quinine (0.5 mM) was added to the 20% alcohol solution, and drinking was assessed during a 24-hour period. In control mice, quinine adulteration suppressed alcohol drinking by 40 ± 8% (Fig. 5J). In iSPN-Drd2HET mice with a high D1/D2 receptor ratio, the same quinine adulteration produced 27 ± 10% suppression in mean lick behavior. Looking at the individual mouse responses, alcohol adulteration left 40% of iSPN-Dd2HET mice consuming as much alcohol as before or more, while only 9% of littermate control mice continued drinking (Fig. 5K, Supplemental Figure 7H, I). Both sexes are represented in the adulteration-resistant group: for iSPN-Drd2HET mice 66% were male and 33% females (4 males, 2 females); for the control group, a single mouse was male. Altogether, while there was no difference in total alcohol consumption, a larger proportion of mice with a high D1/D2 receptor ratio displayed punishment-resistant alcohol drinking compared to controls.

Collectively, these experiments strongly suggest a link between the cellular and circuit mechanisms underlying the stimulant and anxiolytic effects of alcohol in mice. Alterations in the balance of D1/D2 receptor expression and availability in the dorsal striatum generates a risk-aversion phenotype that renders mice more sensitive to the anxiolytic properties of alcohol, which we propose might confer an increased vulnerability for punishment-resistant drinking.

## Discussion

The study, conducted in rodents, aimed to explore the causal relationship and underlying mechanisms between anxiety and alcohol use disorders, observed clinically. Using behavioral and neural manipulations not feasible in humans, we investigated factors regulating alcohol's anxiolytic potency upon initial exposure and its implications for alcohol preference and punishment-resistant consumption. Inbred mice exhibited varying risk-avoidance levels and relief by alcohol in two tests. High-risk avoidance mice showed greater alcohol relief, indicating a link between preexisting risk-avoidance phenotype and sensitivity to alcohol's anxiolytic effects.

We interpreted increased time in "risk" zones post-alcohol administration as evidence of its anxiolytic effects. To differentiate between increased exploration and heightened motor activity, we conducted two experiments. First, the dose of 1.2 g/kg alcohol used in the avoid-explore tests did not increase locomotor activity, unlike a higher dose of 2 g/kg. Second, cocaine, a non-anxiolytic stimulant, increased speed but not time in open zones. These findings argue against alcohol relief from risk avoidance being driven only by enhanced motor output.

Alcohol's relief of risk-avoidance phenotype in C57BL/6J mice relied on dopamine D1R activation as it was blocked by a D1-like antagonist and reduced by lowering D1R in the dorsal striatum. Alcohol induced the activation of D1R-expressing striatal neurons, which were recruited during exploration of the EZM risk zones. These results support the idea that in vivo alcohol-induced elevation of striatal dopamine triggers D1R activation and recruitment of D1R-expressing SPNs, promoting exploration over a risk-avoidance phenotype.

We hypothesized that mice with high D1R expression in the striatum may display stronger relief from alcohol. However, to our surprise, we found that the ratio of Drd1 to Drd2 dopamine receptor mRNA showed the strongest correlation with baseline risk-avoidance phenotype and alcohol relief potency. This novel association of alcohol effects with the ratio, rather than a single receptor subtype, underscores the well-documented 'yin-yang' relationship between the D1 and D2 receptors and the striatal neurons that express them. These G-protein coupled receptors have opposing roles on striatal output and behavior: D1R-expressing neurons promoting and D2R-expressing neurons suppressing striatal output[46]. The receptors also have opposing effects on cAMP signaling, neuronal excitability, and GABA release from the neurons that express them: D1R enhancing and D2R inhibiting. Furthermore, there is crosstalk via axon collaterals and when dopamine activates D2 receptors in SPNs, it suppresses GABA release to disinhibit D1R-expressing SPNs[47,48]. Thus, when dopamine activates D1R and D2R within neurons with opposing behavioral roles, dopamine synergistically promotes striatum-controlled behaviors.

In a collaborative study, we previously demonstrated that optogenetic activation of D2R-expressing striatal neurons promoted risk-aversion behavior, while chemogenetic inhibition had the opposite effect[44]. Combining past and present findings, a more comprehensive model emerges where low expression of D2R causes increased inhibitory output from these NO-GO pathway neurons with D2Rs[45] and promotes risk avoidance phenotype. Simultaneously, high D1R expression enhances alcohol effects by recruiting more D1R-expressing neurons and promoting exploration of risk zones. Thus, a high ratio of striatal D1/D2 receptors explains both phenotypes: heightened risk-aversion at baseline and potent alcohol relief.

We tested the causal relationship between the striatal receptor ratio, the risk-avoidance phenotype, and alcohol relief potency using a genetic manipulation that doubled the D1/D2 ratio in the striatum

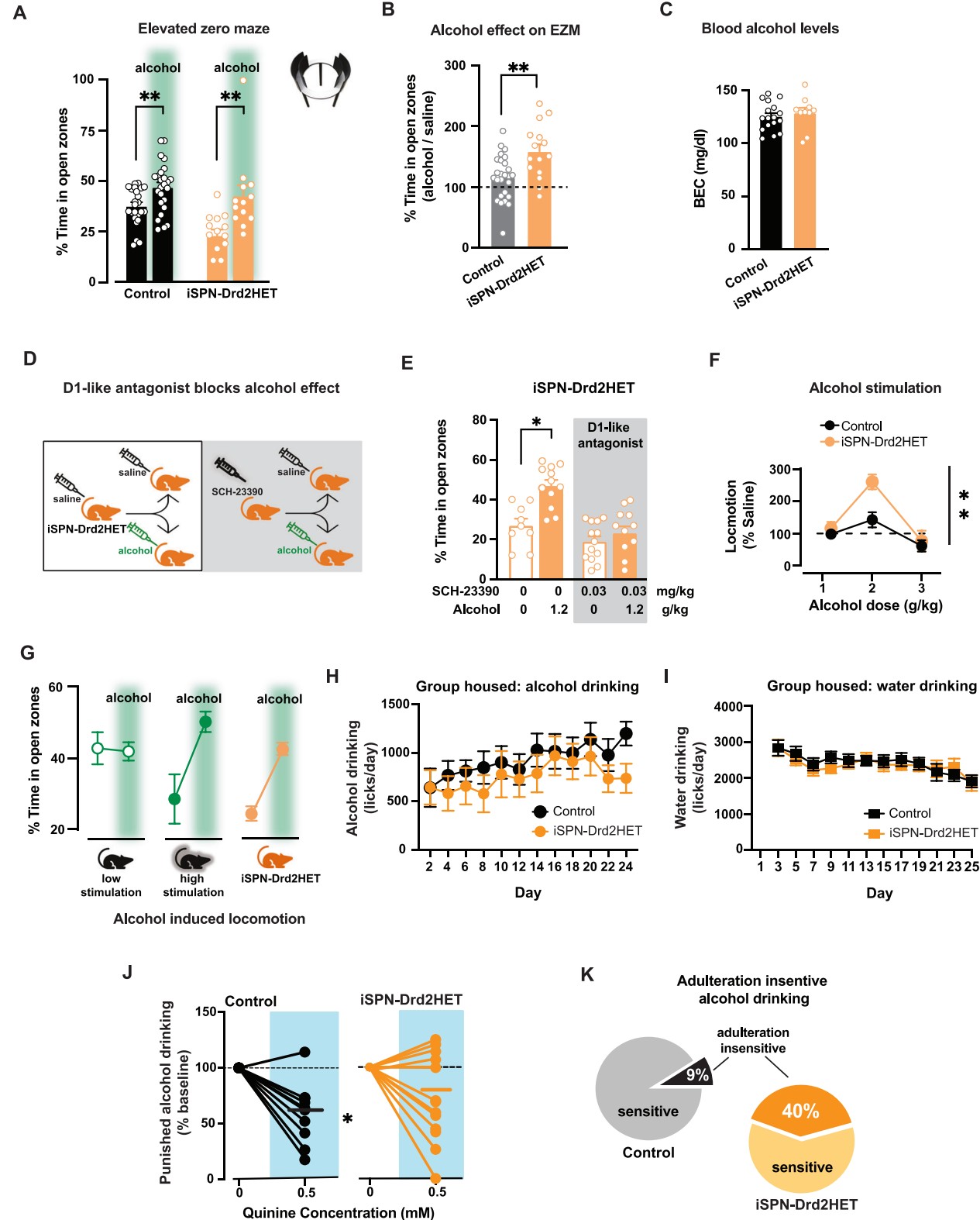

compared to littermate controls. This manipulation produced a risk-avoidance phenotype in mice and promoted alcohol anxiolytic effects, from the very first exposure to alcohol, suggesting these behaviors are the consequence of a preexisting imbalance in striatal dopamine receptor levels.

We hypothesized that the ratio of dopamine receptors in the striatum may be modulated by both internal (e.g. genetic and epigenetic) and external factors (e.g. stressors, social hierarchy[49,50]. The age

of onset is also likely to be an important factor, with changes to the ratio during adulthood having milder implications for the overall brain wiring and embryonic or early-age alterations of ratio having a more profound, long-lasting impact on brain wiring. We speculate that differences in brain wiring among individuals could be associated with phenotypic variation in risk-avoidance and alcohol relief potency. The genetic manipulation used in the study produced a striatum-wide increase in the D1/D2 receptor ratio from early ages, which has clinical

**Fig. 5 | Stronger alcohol relief and aversive-resistant consumption in mice with high ratio of D1 to D2 receptors. A** Time in the EZM open zones for control (black) and iSPN-Drd2HET (orange) mice after receiving either saline or 1.2 g/kg alcohol (green; i.p.; 2-way ANOVA main effect of alcohol $F_{(1,73)} = 2.95$, p = 0.09; control mice: saline vs. alcohol $t_{(48)} = 3.01$, $p = 0.004$; iSPN-Drd2HET: saline vs. alcohol $t_{(25)} = 3.31$, $p = 0.004$, $n = 15,26$ mice/genotype). EZM cartoon is modified BioRender template license Alvarez, V. (2024) BioRender.com/m14d797. **B** Normalized time spent on EZM open zones following alcohol relative to time spent following saline for iSPN-Drd2HET (orange) and controls (black) (t-test $t_{(39)} = 3.31$, $p = 0.002$, $n = 15,26$ mice/genotype). **C** Blood ethanol concentrations (BEC) 10 min after 1.2 g/kg alcohol (i.p.; t-test $t_{(25)} = 0.60$, $p > 0.05$, $n = 10,17$ mice/genotype). **D** iSPN-Drd2HET mice were pretreated with either saline or D1-like antagonist SCH-23390 (0.03 mg/kg; i.p.) then tested on EZM following saline or alcohol (1.2 g/kg; i.p.). **E** iSPN-Drd2HET mice pretreated with saline showed increased time in EZM open zones after alcohol (2-way ANOVA significant interaction SCH-23390 x Alcohol $F_{(3,41)} = 13.5$, $p < 0.0001$; saline versus alcohol: $p < 0.05$, $n = 9,12$ mice/group), while mice pretreated with SCH-23390 did not show increased time in EZM open zones after alcohol compared to saline ($p > 0.05$, $n = 11,13$ mice/group). **F** iSPN-Drd2HET mice show stronger locomotor response at 2 g/kg alcohol than littermate controls (RM 2-way ANOVA main effect of Genotype $F_{(2,62)} = 5.10$, $p = 0.009$; post hoc test $p = 0.011$, $n = 10,23$ mice/genotype). **G** Summary of EZM performance following saline or 1.2 g/kg alcohol (green shade) for control mice with low and high alcohol stimulation and iSPN-Drd2HET mice ($n = 22, 28$ control mice and 15 iSPN-Drd2HET mice). **H** Voluntary alcohol consumption in group-housed, intermittent access, operant paradigm. iSPN-Drd2HET (orange) and littermate controls (black) showed similar consumption of alcohol across sessions (RM 2-way ANOVA no effect of Genotype $F_{(11,241)} = 0.52$, $p > 0.05$, $n = 11,15$ mice/genotype). **I** Water consumption was similar between genotypes on days alcohol was not available (RM 2-way ANOVA no effect of Genotype $F_{(11,263)} = 0.74$, $p > 0.05$, $n = 11,15$ mice/genotype). **J** Alcohol intake was measured following quinine adulteration (0.5 mM), and normalized to intake before adulteration for iSPN-Drd2HET (orange) and control (black; RM 2-way ANOVA significant interaction Genotype x Quinine: $F_{(1,23)} = 4.95$, $p = 0.036$, $n = 11,14$ mice/genotype). **K** Percent mice that drop intake during quinine adulteration (adulteration sensitive) and percent that did not drop (adulteration insensitive); two-tailed binomial test of observed findings (iSPN-Drd2HET) vs expected (control) $p < 0.01$; $n = 11,14$ mice/genotype. Data presented as mean ± SEM, *$p < 0.05$, **$p < 0.01$.

validity given the findings of low D2 receptor availability[51–53] and no changes to D1Rs in AUD positive individuals across the striatum[54], suggesting that D1/D2 receptor ratio might also differ among humans with a history of alcohol abuse. Furthermore, the D2 reduction did not improve with protracted withdrawal, indicating the changes might be an intrinsic characteristic and precede the alcohol exposure[55,56].

Our findings suggest a model in which both dopamine receptor types, D1 and D2, contribute to vulnerability for AUD. The imbalance in the activity of both receptor subtypes emerges as a crucial feature to monitor in assessing vulnerability. These findings complement our previous reports showing functional upregulation of D1R signaling following either low D2R[35] or loss of Lrrk2 function[57], which both led to enhanced acute potency of alcohol, drinking preference and punishment resistant drinking. The receptor imbalance is expected to translate into a pathway imbalance between the direct- and the indirect-projection pathways. The pathway imbalance is hypothesized to have distinctive behavioral and circuit-level implications: high D1/D2 receptor ratio results in low D2R activation and weak suppression of GABAergic transmission from D2 SPNs, which in turn causes a strong lateral inhibition onto D1 SPNs[45,58], promoting risk-avoidance phenotype[44]. When dopamine levels are elevated by alcohol, a high D1/D2 receptor ratio results in strong D1R activation and more potent recruitment of direct-pathway neurons[57] that facilitate risk-taking and exploration behavior.

We propose that the anxiolytic properties of alcohol may contribute to enhancing its reinforcing properties in subjects with unbalanced circuits, increasing the likelihood of alcohol abuse and dependence. In support of this hypothesis, alcohol anxiolytic potency was positively correlated with alcohol drinking levels in wild-type mice. We also used the Intellicage System to assess operant alcohol consumption and the effect of quinine adulteration on alcohol drinking behavior in a group-housed setting. While there was no difference in overall alcohol consumption within the 3 weeks tested, mice with low striatal D2Rs displayed stronger punishment-resistant alcohol drinking, a selective phenotype associated with AUD in humans. Based on findings from this current study, we propose that the critical factor for vulnerability is the imbalance of dopamine receptor availability and function within the striatum rather than the levels of a single dopamine receptor. Furthermore, future focus on alcohol's anxiolytic effects in humans could prove useful as a behavioral biomarker in populations at risk for AUD. Ultimately, this study offers a plausible neuronal mechanism by which enhanced acute effects of alcohol can promote drinking and lead to long-term maladaptive drinking.

## Methods

### Animals

All procedures were performed in accordance with guidelines from the NIAAA, NIMH and Rutgers University Animal Care and Use Committees. All experiments, unless otherwise stated, used male and female mice in near equal proportions (C57BL/6J background, 8–10 weeks old at time of behavioral testing) that were group-housed on 12 h:12 h light cycle (6:30 on, 18:30 off) with *ad libitum* access to standard rodent chow and water and dry-bulb temperature of 20–26 °C and relative humidity in range of 30–70%. All mouse lines used for breeding are commercially available. Mice with a knockdown (iSPN-Drd2HET) or knockout (iSPN-Drd2KO) of dopamine D2 receptors on striatal projection neurons (Adora2a-Cre$^{+/-}$; Drd2$^{loxP/wt}$ and Adora2a-Cre$^{+/-}$; Drd2$^{loxP/loxP}$, respectively) were generated by crossing Drd2$^{loxP/loxP}$ mice (B6.129S4(FVB)-*Drd2$^{tm1.1Mrub}$*/J, IMSR_JAX:020631), which carry the conditional allele for Drd2, with Adora2a-Cre$^{+/-}$ mice (B6.FVB(Cg)-Tg(Adora2a-cre)KG139Gsat/Mmucd, MMRRC ID: 36158) which express Cre recombinase under the adenosine 2a receptor promoter. For all experiments, Cre negative Drd2$^{loxP/loxP}$ or Drd2$^{loxP/wt}$ littermates were used as controls. Drd1$^{loxP/loxP}$ mice (*Drd1$^{tm1Jcd}$*/J; IMSR_JAX:002322) were used in Cre-dependent viral vector injection studies. GCaMP fiber photometry studies utilized Drd1a-cre mice (B6.FVB(Cg)-Tg(Drd1-cre) EY262Gsat/Mmucd). All mice were genotyped at weaning using real-time PCR with their respective probes by Transnetyx (Cordova, TN).

### Behavior

For all experiments, experimenters were blinded to group allocation during data collection and/or analysis. Sample sizes were determined using power analyses; standard deviation estimations were based on previous data from our groups or published in the citations provided. All testing was done during the light period and mice were habituated to the testing room for 1 h prior to experimentation. All behavior was recorded using a GoPro camera and EthoVision XT 10 software was used to analyze recordings, unless otherwise stated. Experimenter was blind to the genotype during behavior testing and data analysis.

### Elevated zero maze

The elevated zero maze (EZM) consisted of a ring-shaped runway (50 cm inner diameter, 5 cm lane width, 50 cm off the ground) with equal amounts of space devoted to open and walled quadrants (15 cm wall height). In Figs. 1, 2, 5, following one week of 3x saline injections (10 mL/kg, i.p.) for habituation before testing. For experiments using a single EZM exposure, mice were injected with ethanol

(1.2 g/kg,10 mL/kg, i.p.) or saline before being immediately placed on an open quadrant of the elevated zero maze. Mice were allowed to freely explore for 10 min. For the repeated EZM tests, mice received injections of either saline (10 mL/kg, i.p.) or ethanol (1.2 g/kg,10 mL/kg, i.p.; order counterbalanced) in two tests conducted one week apart. Factors such as time of day and lighting were kept as consistent as possible. In Supplementary Fig. 3G, mice were injected with cocaine (10 mk/kg, 10 mL/kg, i.p.) and placed on the EZM. In Fig. 4, transgenic mice were placed directly on the open quadrant of the elevated zero maze for 10 min and in Fig. 5 mice were pretreated with either SCH-23390 (0.03 mg/kg) or saline at time zero, and 15 min later injected with ethanol (1.2 g/kg) or saline and immediately placed on an open quadrant of the elevated zero maze for 10 min. Noldus EthoVision XT 10 software was used to sum the total time on open quadrants. The semiautomatic analysis of time spent in open zones was checked by researchers blind to the experimental conditions and found to be accurate. However, the mouse location while in the closed zones was not accurately assessed from the video in most cases, which precluded any measures of distance travel in the EZM.

### Open field
Mice were treated with either ethanol (1.2 g/kg) or saline. Following treatment mice were placed in the center of a novel box (30 × 40 × 40 cm) for free exploration (30 min). Noldus EthoVision XT 10 software was used to determine the time spent in the center of the box and overall locomotion.

### Light-dark box
A novel 30 × 40 × 40 cm box was divided into open (light) and walled/covered (dark) zones. Two-thirds of the box comprised the light zone, while one-third of the box comprised the dark zone. Uninhibited passage between zones was allowed. Light levels in the open zone measured ~95 lux. Mice were administered ethanol (1.2 g/kg) or saline. Following treatment, mice were placed in the center of the open zone for free exploration (12 min). Noldus EthoVision XT 10 software was used to determine the total time in light versus dark zones.

### Latency to feed
Mice were food deprived for 12 h, then placed in a novel open field (30 × 40 × 40 cm) under high illumination (400–750 lux) with food in the center. Videos were analyzed to determine the amount of time it took the animal to approach the food.

### Alcohol stimulatory response
Mice were placed in polycarbonate chambers (20 cm H × 17 cm W × 28 cm D) equipped with infrared photobeam detectors (Columbus Instruments). Beam breaks were recorded 1 h for habituation to locomotor chambers before mice received i.p. injections and beam breaks were recorded for an additional hour. Mice received saline 10 mL/kg, i.p. for days 1–3 to habituate to handling and injections. Subsequent days, during the testing phase, mice were administered either saline or ethanol (2.0 g/kg for Figs. 1, 1.0, 2.0 and 3.0 g/kg in Fig. 5; i.p.) using a counterbalanced design.

### Sucrose preference test (anhedonia)
Mice were singly housed and had *ad libitum* access to rodent chow, water, and 1% sucrose (w/v). Water and sucrose were presented in glass tubes (25 × 3 × 100 mm, Pyrex) fitted with straight, open-tipped metal sippers. Glass tubes were weighed every 24 h to determine intake, and the position of the water and sucrose tubes were switched. Procedure was repeated daily for 4 days. Preference was calculated as the ratio of the volume of sucrose solution consumed divided by the total volume consumed in 24 h and averaged for the final 3 days.

### Social-operant alcohol drinking
The Intellicage testing system (TSE) was used to group house mice, while still collecting individual data on alcohol intake. One week prior to Intellicage testing, mice (7–11 weeks) were briefly anesthetized and implanted with a subcutaneous radiofrequency identification transponder (RFID) chip, supplied by the manufacturer. Chips were placed in the dorsocervical region, not impeding the animals' movement or locomotion. Same-sex groups of mice were housed in the Intellicage under reverse light cycle (8–14 mice/cage). The testing apparatus consists of a large cage with 4 corner chambers, each equipped with an RFID antenna and accessible through an open tunnel. Each corner chamber contains two doors that are computer program operated, and control access to the fluid bottles. For a visit to be registered, and the mouse given access to the fluid bottles, the RFID chip must be scanned in conjunction with a temperature sensor detecting heat. Nose pokes and licks are detected by sensors in the nose ports and bottle spouts, respectively. Data was extracted using the TSE analyzer software and processed in R. For each task, a description of the paradigm is described below (see manufacturer handbook for more specifications).

**Water training.** At the start of the experiment, the mice were given free access to the corners until the first visit closed all the doors. Water access was given with a FR1 schedule. Doors would open and allow 5 sec access to water. To regain access to the water, the mice would have to leave the corner and return, registering a separate visit for every time they gained access to the water. The nose pokes per visit were recorded and the visit registered as a success trial. Once an individual mouse reached a 60% success rate (60% or higher success trials out of total trials), the program moved to a FR 3 schedule for the remainder of the program duration. A successful FR 3 was 3 pokes on the same side within 2 s. This program was run for approximately a week until all mice successfully achieved FR 3.

**Intermittent alcohol intake.** Here, two corners were designated as "alcohol corners", where mice would only have access to alcohol (20% ethanol; v/v) while the other two corners contained water. All doors for fluid access opened with a FR3 nose poke schedule. Ethanol was prepared by diluting 95% alcohol (190 proof, Deacon Labs) in tap water. The alcohol corners were only accessible every other day. Aside from weekly cage changes and single day of progressive ratio testing, the program was run for approximately six weeks total.

**Alcohol adulteration.** This program was identical to the above "Intermittent Alcohol" program, with the only difference being the alcohol bottles administered were adulterated with 0.5 mM quinine. The program was run for 3 days, with 2 days being water-only access and 1 day being alcohol and water access. Any mice not drinking alcohol in the "Intermittent Alcohol" phase were excluded from analysis in the "Alcohol Adulteration" phase (1 mouse).

**Data analysis.** The IntelliCage data was analyzed with the TSE provided analyzer program and the data exported into R, where they were cleaned, summarized, and analyzed using publicly available packages from the tidyverse.

### Stereotaxic viral injection
Drd1$^{loxP/loxP}$ mice were placed in a stereotaxic frame under isoflurane anesthesia and bilaterally infused with a viral vector expressing Cre (AAV9.CMV.HI.eGFP-Cre.WPRE.SV40; UPenn) or a control vector (AAV2/9.CB7.CI.EGFP.RBG, UPenn) into the dorsal striatum. Viral injections (200 nL/injection) were delivered at a rate of 100 nL/min using a Nanoject II (Drummond Scientific). The injection pipette was kept in place for 4 min following the infusion to promote proper diffusion and prevent backflow. Stereotaxic coordinates were (mm from

bregma): + 0.5, + 1.0 AP, ± 1.5, ± 1.75 ML, − 2.75 DV. Behavioral experiments began 4 weeks after viral injections. Viral expression was confirmed by fluorescence visualization and qPCR.

### Fiber photometry

To optically monitor calcium using in vivo fiber photometry, a viral vector encoding the green fluorescent calcium indicator, GCaMP6s (AAV5-Syn.flex.GCaMP6s.WPRE.SV40; $7 \times 10^{12}$ vg/mL; Addgene plasmid: 100845), was unilaterally infused into the dorsal striatum of Drd1a-Cre mice. Stereotaxic coordinates were (mm from bregma): +0.5, +1.0 AP, ± 1.5, ± 1.75 ML, −2.75 DV. Immediately following viral vector injections, optic-fiber cannula (200 µm; 0.48 NA; M3 thread titanium receptacle; Doric Lenses) were implanted in the dorsomedial striatum over the GCaMP6s injection coordinates (same A/P and M/L as GCaMP6s injection described above, D/V point was placed at −2.6 mm). Fibers were secured to the skull with Metabond (Parkell) and dental cement. Miniature screws were placed in the skull posterior and lateral to fiber to support cementing. Mice were single housed after fiber implant surgery to prevent damage to the implants.

**Signal measurement.** To measure signals, 470 nm and 405 nm fiber coupled LEDs (M470F3 & M405FP1, Thorlabs) were connected to a fluorescence filter minicube (Doric), measured emission light, bandpass filtered through the minicube, and measured with a femtowatt photoreceiver (Newport, model 2151). The signal processor demodulated the detector signal producing separate emission signals for 470 nm excitation and 405 nm excitation (isosbestic control). Sampling rate of the demodulated signals was 1017 z, lowpass filtered at 20hz. 470 nm and 405 nm LED powers were ~100 µW and ~40 µW, respectively, measured at the patch cord tip. Light powers were kept consistent across animals and sessions. Patch cords were bleached overnight before testing with continuous 470 nm light to reduce cable autofluorescence.

### Behavioral testing

**Habituation.** Four or more weeks following fiber placement surgery, mice are acclimated to handling and being attached to the fiber tether. For three days, each mouse was moved into the behavioral room and allowed to acclimate for ~30 min, mice were handled, plugged into the photometry system, and allowed to move around an open arena for ~30 min.

**Elevated zero maze.** On experimental Day 1, similar to the habituation days, mice were acclimated to the behavioral room for ~30 min, plugged in to the photometry system and allowed the mouse to move around for ~10 min in a clean home cage. Each mouse was then placed on an elevated zero maze for 30 min.

**Alcohol exposure.** On experimental days 2 and 3, mice were acclimated to the behavioral room for ~30 min and plugged into the photometry system. The mouse was allowed to move around in a clean home cage for 10 min, before receiving an injection of saline (10 mL/kg, i.p.) and placed in the open field for 30 min. On testing days 4 and 5, the same procedure was followed, however, 30 min following the saline administration mice received an injection of either 1.2 or 2 g/kg ethanol (i.p., counterbalanced design) and were again placed in the open field for a 30-minute recording session.

**Data analysis.** Data was analyzed with custom Python scripts based, in part, on the GuPPy package[59]. Both the 405 and 470 nm channels were filtered with a $6^{th}$ order / 6 Hz Butterworth lowpass filter before the 405 nm channel was fit to the 470 nm channel via least-squares regression. To calculate ΔF/F, the difference between the fitted 405 and 470 nm channels was divided by the fitted 405 nm channel (Eq. 1).

Post-injection area under the curve (AUC) was calculated in one-minute bins and normalized to a five-minute pre-injection baseline. Event frequency was calculated using published methods[59]. The light-dark ΔF/F was cut into 5-minute sections over which a median signal was calculated. A second median was calculated by filtering out any signal greater than two standard deviations from the initial median. Events were identified as any peak greater than 3 standard deviations from the second filtered median. Frequency was then calculated over 20 s bins.

$$\frac{\triangle F}{F} = \frac{signal - fitted\ control}{fitted\ control} \quad (1)$$

**Euthanasia.** Once testing was completed, mice were deeply anesthetized with isoflurane and perfused with 4% PFA. The brains were collected for histology to confirm fiber location and viral expression.

### Physiological Measurements

**Serum corticosterone levels.** Blood samples were rapidly collected by nicking the tail vein with a razor blade and ~50 µL of blood was collected in heparinized capillary tubes. Tubes were centrifuged for 5 min, and plasma was isolated for later analysis for corticosterone for Eliza (CrystalChem, #80556).

**Blood ethanol concentration (BEC) Measurements.** Mice were injected with 1.2 g/kg alcohol and blood samples were collected 10 min later. The tail vein was nicked with a razor blade and 15–50 µL of blood was collected in heparinized capillary tubes. Tubes were centrifuged for 5 min, and plasma was isolated. BECs in plasma samples were analyzed in duplicate using the Analox analyzer GM7 MicroStat (Analox Instruments, Lunenburg, MA).

**Quantitative polymerase chain reaction.** Mice were anesthetized with isoflurane and decapitated. Brains were removed, and the striatum was dissected on ice using a 1 mm coronal matrix, placed in RNAlater, homogenized, and total RNA was purified using RNeasy Plus Mini kit (QIAGEN). cDNA was synthesized using iScript Reverse Transcription Supermix (Biorad). Actb (Mm01205647), Drd2 (Mm00438541_m1), and Drd1 (Mm02620146_s1) TaqMan Gene Expression Assays (Applied Biosystems) were used to determine relative mRNA expression of the endogenous control gene β-actin, dopamine D2R and dopamine D1R, respectively. Samples were run in triplicate and in parallel with negative controls using the StepOnePlus Real-Time PCR system (Applied Biosystems). The cycling conditions were: initial hold at 95 °C (20 s), 40 cycles of 95 °C (1 s) and 60 °C (20 s). Relative Drd1 and Drd2 mRNA levels were calculated using the ΔΔCt method. For the Drd1 knockdown experiments of Fig. 2, the viral vector injection was localized to dorsomedial striatum and tissue collected 4–6 weeks later and after behavioral testing. Due to the need for prompt dissection of tissue under RNAase free conditions to prevent mRNA degradation, it was not possible to confirm the infection site and spread by assessing fluorescence.

**Autoradiography.** D1 and D2 receptor expression were measured in the striatum of mice using autoradiography. Whole brains were flash-frozen in 2-methylbutane for later sectioning (20 µm) on a Cryostat (Leica) and thaw mounted onto ethanol-washed glass slides. Slides were preincubated (10 min, room temperature) in washing buffer (50 mM Tris-HCl, pH 7.4 with 120 mM NaCl, 1 mM MgCl2, 5 mM KCl, 2 mM CaCl2), then transferred to incubation buffer (60 min, RT; Tris-HCl washing buffer) containing either [3H]raclopride (4 nM, 65 Ci/mmol, PerkinElmer) or a combination of [3H]SCH-23390 (2.5 nM, 83.6 Ci/mmol, PerkinElmer) with

Ketanserin (40 nM, Tocris) in order to block off-target binding of SCH-23390 to serotonergic receptors. Another set of slides containing consecutive sections was incubated in the same conditions in the presence of either butaclamol (10 μM, Tocris) or SCH-23390 (10 μM, Tocris), respectively, to determine nonspecific binding. Slides were washed in an ice-cold incubation buffer twice and rinsed in ice-cold water. Then the slides were dried overnight, placed in a Hypercassette (Amersham Biosciences), and covered with a BAS-TR2025 Storage Phosphor Screen (Fujifilm). A slide containing Carbon-14 Standards (American Radiolabeled Chemicals Inc.) was exposed simultaneously for quantitative radiometric analysis. The slides were exposed to the screen for 10 days and imaged using a phosphorimager (Typhoon FLA 7000; GE Healthcare). Images were calibrated and analyzed using ImageJ 1.51j8 (NIH). Regions of interest (ROI; 10 to 16 ROI per region and animal) were drawn freehand based on neuroanatomical landmarks and quantified by densitometry. The non-specific binding (nCi/g) in each ROI was subtracted from total binding to calculate specific binding (nCi/g), and the global specific binding in the dorsolateral striatum for wild-type animals was used to normalize specific binding values within each individual experiment.

**Fast-scan cyclic voltammetry.** Mice were anesthetized with isoflurane and killed by decapitation. Brains were quickly removed, mounted, and sliced in a vibratome (VT-1200S; Leica) in an ice-cold cutting solution containing the following (in mM): 225 sucrose, 13.9 NaCl, 26.2 $NaHCO_3$, 1 $NaH_2PO_4$, 1.25 glucose, 2.5 KCl, 0.1 $CaCl_2$, 4.9 $MgCl_2$, and 3 kynurenic acid. The obtained coronal slices (240 μm) were recovered for 20 min at 33 °C in artificial CSF (ACSF; in mM: 124 NaCl, 1 $NaH_2PO_4$, 2.5 KCl, 1.3 $MgCl_2$, 2.5 $CaCl_2$, 20 glucose, 26.2 $NaHCO_3$, and 0.4 ascorbic acid) and maintained at room temperature until recordings. For recordings, slices were submerged in a chamber with continuous perfusion (2 ml/min) of ACSF and kept at 32 °C using an in-line heater (Harvard Apparatus). Fast-scan cyclic voltammetry (FSCV) was performed in the dorsomedial striatum. Cylindrical carbon-fiber electrodes were prepared with T650 fibers (6 μm diameter, ~150 μm of exposed fiber) inserted into a glass pipette. The carbon-fiber electrode was held at −0.4 V versus Ag/AgCl and a triangular voltage ramp ( − 0.4 to +1.2 and back to −0.4 V at 0.4 V/ms) was delivered every 100 ms. Dopamine transients were electrically evoked, a glass pipette filled with ACSF was placed near the tip of the carbon fiber (~100–200 μm), and a rectangular pulse (0.2 ms) was applied every 2 min. The amplitude of the current pulse (100–250 μA) was adjusted to use the minimal current needed to generate a maximal and stable response. Data were collected with a retrofit head stage (CB-7B/EC with 5 MΩ resistor) using a Multiclamp 700B amplifier after low-pass filtering at 10 kHz and digitized at 100 kHz using pClamp10 software (all from Molecular Devices). For analysis, baseline voltammograms before stimulation were averaged and subtracted from the voltammograms during and after stimulation, and transients were calculated from the oxidation peak region using custom-written analysis software in Igor Pro (WaveMetrics). The current peak amplitude of the evoked dopamine transients were converted to dopamine concentration according to the post experimental calibration of the carbon-fiber electrodes with dopamine (1–3 μM) applied locally through a glass pipette in the recording chamber. Transients were compared before and after bath application of 80 mM ethanol.

**Drugs.** Ethanol (Decon Laboratories, 190 proof) was dissolved in water or saline at 20% (v/v). Quinine (Sigma-Aldrich) was dissolved in the 20% alcohol solution, according to the experimental conditions. Cocaine HCl (National Institute on Drug Abuse), and SCH-23390 (ab120597) were dissolved in saline. Intraperitoneal (i.p.) drug administration was delivered at 10 ml/kg body weight.

**Quantification and statistical analysis.** Analyzes were performed in Prism 9 (GraphPad). All statistical tests run were two-sided and data is presented in the figure legends. Data comparing two groups (elevated zero maze, light-dark box, stimulatory test, blood ethanol concentrations) were compared with an unpaired Student's t-test. Data making three or more comparisons were analyzed using two-way ANOVA, with the addition of repeated-measures (RM) or a mixed-effects model, as appropriate. For RM-ANOVA, sphericity was assessed with Mauchly's test, and Greenhouse–Geisser correction was applied when appropriate. Sidak or Tukey tests were used for multiple comparisons. Significant main effects or interactions were followed-up with pairwise tests corrected for multiple comparisons. Paired or independent sample t tests were used to analyze the remainder of data. Results were considered significant at an alpha ≤ 0.05. All data are presented as mean ± SEM and individual animal data also showed whenever possible. Sample sizes were chosen based on previous research employing similar approaches and are sufficient for detecting strong effect sizes while using the minimal number of animals.

### Reporting summary
Further information on research design is available in the Nature Portfolio Reporting Summary linked to this article.

### Data availability
The datasets generated during the current study are available on Mendeley Data, V1, https://doi.org/10.17632/82zj46jr97.1 https://data.mendeley.com/datasets/82zj46jr97/1 or from the corresponding author on reasonable request.

### Code availability
No original code was generated. Publicly available code for the analysis of fiber-photometry data displayed in Fig. 2k–r with minor custom modifications (modified code is made available in the data repository). Any additional information required to reanalyze the data reported in this paper is available from the lead contact upon request.

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

## Acknowledgements

This study was supported by ZIA-AA000421 (VAA), ZIA-MH002987 (VAA); NIH K99/R00AA027750 (MEB) and ZIA-DA000069 (MM). In memory of Patrick Hong (1998–2022) who made significant contributions to this work.

**Declaration of generative AI and AI-assisted technologies in the writing process.** During the preparation of this work the author(s) used ChatGPT to provide grammar suggestions. After using this tool/service, the author(s) reviewed and edited the content as needed and take(s) full responsibility for the content of the publication.

## Author contributions

Conceptualization and Methodology: M.E.B., D.A.B., M.M. and V.A.A. Investigation, Validation and Analysis: M.E.B., M.J.S., L.G.A., E.V., H.B.K., C.E.T., M.B., H.C.G., R.B., I.M.K., R.R., P.H., E.M.M., J.B. and D.A.B. Writing: M.E.B. and V.A.A. Funding, Resources, and Supervision: V.A.A., M.E.B. and M.M.

## Funding

## Competing interests

M.M. has received research funding from AstraZeneca, Redpin Therapeutics, Attune Neuroscience, and Dompé farmaceutici. All other authors report no competing interest.
