## [Transparent Peer Review file · Nature Communications]

Preexisting risk-avoidance and enhanced alcohol relief are driven by imbalance of the striatal dopamine receptors in mice

Corresponding Author: Dr Veronica Alvarez

Version 0:

Reviewer comments:

Reviewer #1

(Remarks to the Author)

The manuscript by Bocarsly et al., Shift of balance in dopamine D1 and D2 receptors enhances the anxiolytic potency of alcohol and promotes punishment-insensitive drinking in mice, attempts to explore the causal relationship between anxiety and alcohol in an inbred mouse model. They show that a low dose of alcohol enhances anxiolytic-like behaviors as compared to saline controlled mice, and that this is in part mediated by striatal D1R-expressing projection neurons. The authors use fiber photometry to demonstrate D1R-expressing neurons are recruited during these behavioral assays. The use of striatal specific *Drd1a* knockdown mice, and *Drd2Het/KO* mice, is interesting. While the overall hypothesis is interesting, I don't believe the data adequately tests it and much additional work is needed to support the conclusions.

Major Points

(1) The hypothesis that basal anxiety-like states predicted drinking, or that alcohol provide relief for anxiety-like behaviors, is not tested in this design. The shifts in overall population behavior in Figure 1 do not get at this point. Showing that these behaviors are correlated with each other (Supp Figure 1) also does not answer this. While I appreciate the difficulty of using non-repeatable assays for measuring anxiety-like behaviors, running saline-exposed mice and alcohol-exposed mice in parallel simply does not answer this question. The authors would need to include experiments that actually address this point (some options):

- a. repeatable anxiety assays (eg marble burying) to demonstrate that basal anxiety predicts the response to ethanol
- b. CORT levels at baseline, to show basal anxiety correlated with EtOH-governed responses
- c. Pretest in one assay (eg EZM), sort the mice based on phenotype, and show predictive validity for post-EtOH effects in another assay (not just correlations)

(2) Similar to point 1 - a key argument in the abstract is that the anxiolytic effects of alcohol predispose individuals to punishment-resistant drinking. There is not nearly enough evidence in the manuscript to make this claim. The D2R mutants happen to exhibit both alcohol-induced anxiolysis and punishment-resistant drinking phenotypes. To claim that anxiolytic effects make them prone to punishment-resistant drinking, the authors would have to separately show that anxiety-resistant mice generally show punishment sensitive drinking right, and if you manipulate the anxiety phenotype you should be able to manipulate the drinking. Otherwise the authors simply can't say this association is causal.

(3) Figure 1 uses 1.2 g/kg alcohol, which only modestly recruited striatal D1R neurons (and thus the authors shift to include 2 g/kg with the fiber photometry). It would be interesting to know if 2 g/kg was more predictive in all of their work.

(4) I don't believe that C57Bl/6J mice are the appropriate strain for the overall study. These mice are inbred and in reality do not show much range in anxiety-like phenotypes, which is the point of the study.

(5) The authors indicate mice were used ranging from initial PND 45-120. This earlier range is well within adolescence, a known time period of altered response to alcohol and with differing anxiety-like phenotypes. Do the authors see any differences when accounting for age?

(6) Figure 1J – wouldn't the better experiment be to show that saline performance on the EZM predicted drinking behavior in

the two bottle choice?

Reviewer #2

(Remarks to the Author)

I express first my condolences for the sad loss of Patrick Hong and I commend you for honoring them in your manuscript.

In this paper, the investigators are attempting to answer the question on whether anxiety-like phenotypes precede problematic alcohol use by using a mouse model of behavioral responses to alcohol after testing for baseline anxiety-like behaviors. They utilize an impressive multidisciplinary approach that focuses on the expression of dopamine receptors in the striatum. They conclude that the balance of D1 dopamine receptor expression and D2 dopamine receptor expression in different populations of spiny projection neurons (also known as medium spiny neurons, MSNs) accounts for the anxiety-like behavior and the anxiolytic response to alcohol. The study is exciting and important. I have a number of concerns and critiques that I will present below. In the absence of line numbers, I will refer to page numbers and paragraph numbers where I need to comment on a specific statement.

1. The statement (page 2, paragraph 2) that rodents consume more alcohol after a period of abstinence is not necessarily true. There are many studies that show that this is not universally true. In addition, the argument that such a behavior is due to anxiety does not really follow.
2. It was good to include both males and females in the study. It is never made clear though whether there were equal numbers of males and females of every genotype in every experiment. An imbalance in males and females could skew the data.
3. Considering that there were many statements about individual variability, I found it surprising that there were very few, if any analyses that specifically determined within-animal correlations.
4. There were multiple instances of referring to correlations of 0.4 to 0.6 as strong correlations. I know this is a bit of a semantics argument, but I would not consider correlations around 0.5 to be strong, but rather moderate.
5. I lost count of how many times there were errors in which panels of which figures were being referred to in the text. Supplemental Figure 3 was never referenced. Supplemental Figure 4 was only ever referenced in the methods. It went straight from Supplemental Figure 2 to Supplemental Figure 5. I am not sure if these data were even discussed.
6. The units for Figure 1A and 1B in the text do not match the units in the Figure itself.
7. It was not clear how the doses of alcohol were chosen and it wasn't considered that individual variability or genotype differences might represent shifts in the dose-response relationships rather than an absence or presence of a response altogether.
8. Shifts in distribution were provided/described a few times, but there were not analyses to show a significance in these shifts.
9. Locomotion data (Figure 1) were provided for the open field, but not for the EZM or the light dark box which was inconsistent.
10. An increase in alcohol consumption was assessed as a corollary of anxiety-like phenotype, but total fluid consumption was not provided. Polydipsia is an alternative explanation rather than enhanced alcohol consumption.
11. What was the rationale for performing photometry in the DMS rather than some other subregion of the striatum (DLS, VMS, ventral striatum)?
12. It was not explained what the relevance of the decrease in the AUC of the calcium signal was following alcohol treatment.
13. What is the explanation for why there was only a 55% drop in *Drd1a* levels with the genetic approach? Why was it not higher? I don't expect 100%, but 55% seems low.
14. I understand the difficulties of genetic manipulations of D2R expression, but there is a bit of a disconnect between the DMS-focused D1R experiments and the breeding strategy for manipulating D2R expression that produces a loss of D2R striatum-wide. This should be addressed as a limitation of their study.
15. The D2R expression manipulations all decreased locomotion. How did the investigators disambiguate the differences in time spent in different areas from just reduced motor activity?
16. The lack of change in the full D2R KO in Figure 3E is ignored.
17. I have major concerns with the interpretation of the quinine consumption experiments, based on how the data were presented. First, the group sizes are very different 6 vs. 13 animals. Only 5 of the 13 mutants maintained drinking in the presence of quinine. The data are all baselined, so there is no way to determine if more alcohol consumption led to greater quinine-resistance or, considering that of the 13 mutant mice there were a lot that consumed very little alcohol, that when the data are normalized, there isn't much a decrease in consumption because they were already drinking very little alcohol. The groups should be better balanced and the data presented in a more clear fashion.
18. Page 9, paragraph 4 states that there was an increase in D1:D2 binding ratio specifically in the DMS. That is absolutely not what the data show.
19. It isn't surprising that there is a change in the D1:D2 ratio when there is a substantial drop in D2 expression. I don't think that the investigators presented data to sufficiently prove that the ratio of the expression of the two receptors is what determines anxiolytic baseline behavior and responses to alcohol. I would ask that they use another method to shift the balance of the two receptors, perhaps by selectively overexpressing one of the two types of receptors in a non-knockout mouse to shift the balance when there is not an absence of the other receptor.
20. Page 11, paragraph 3 proposes a model whereby a "jolt of dopamine" produces the behaviors that are imbalanced because of differing ratios of D1:D2. They performed one experiment with cocaine in a wild type mouse (Figure 1L) to show that this had no effect on anxiolytic behavior. If they want to argue that the imbalance of receptor responses leads to different responses to "jolts" of dopamine release, what better way than to test the imbalance with another test of cocaine?

21. The final paragraph of the discussion section argues that it is the ratio of D1:D2 in the DMS that produces the behavioral outcomes. In no way do their experiments support a specific role for the DMS in any of the behaviors tested.
22. The age range at the start of experimentation was very wide. Are there concerns that different ages of mice will lead to different behavioral outcomes?
23. There is no statement regarding whether the data were tested for normality before proceeding to use tests that are for normally-distributed data.
24. The supplemental checklist indicates that blinding was used. For the average reader who will not look at the checklist, it would be good to include this statement in the methods section.
25. What does Supplemental Figure 5B show that Figure 5C does not already show? Individual values?
26. I found the organization/presentation of Figure 1I to be very confusing.

Reviewer #3

(Remarks to the Author)

The manuscript by Bocarsly et al. is an intriguing examination of the roles of D1R and D2R in the dorsomedial striatum, and the balance between them, on the acute effects of alcohol on avoidance behavior and locomotion. This is an area of investigation in the alcohol research field that has been under-explored in the literature, and there is potential for this work to contribute significantly to the field. However, there is some lack of clarity in the presentation and reporting that make it difficult to understand the actual results and evaluate the interpretation of the data.

One broad issue is that there is no evidence that anxiolytic effects of alcohol (and manipulations altering these) are not due to locomotion. This stems from several related questions/issues included below.

1) Lack of clarity about the specific experimental conditions within each figure/panel and inconsistency across them. Was all of the behavior done on the same mice or different groups of mice? Can the authors provide experimental timelines at the top of each fig/relevant group of panels?

2) What is the point of the distribution graphs for avoidance behavior, given that the distributions are fairly obvious in the bar graphs above them? They generally look like the gaussian distribution shifts, with typical large variability within group and a high degree in overlap of distribution across groups, suggesting a general effect across mice, however the authors argue that some mice are more susceptible than others to alcohol-ON anxiolysis.

3) What is the alcohol stimulatory score? Is it a difference in locomotion between saline and alcohol within each mouse or just the alcohol ON locomotion? If the latter, this should not be referred to as the stimulatory effect (implying within-mouse) because that is not what it is showing. Similarly, the authors frequently refer to the anxiolytic effect of alcohol as the alcohol-ON avoidance behavior, but they generally do not show that it is anxiolytic within-mouse (except Supp Fig 4).

4) Relatedly, why is the alcohol stimulatory score used as a proxy for the anxiolytic effects? Why isn't an anxiolytic score as in Supp Fig 4 used throughout to more directly show alcohol-induced anxiolysis?

This issue permeates the other analyses and data interpretation. For example, they use the alcohol stimulatory score in correlations with other measures like basal avoidance and interpret that as high basal avoidance being related to alcohol-induced anxiolysis. Is there any evidence for this relationship that does not depend on the locomotion? For example, is alcohol ON avoidance correlated with basal avoidance? Same for alcohol drinking phenotype (as in 1J)-- how does alcohol drinking compare to basal (saline) anxiety?

5) The authors do not seem to present the locomotor data from the anxiety assays, nor examine the relationship between that locomotion and avoidance behavior on tests. As there is often a locomotor phenotype at baseline or in response to alcohol, differences in locomotion at basal/alcohol conditions may be a confound for interpretation of anxiolytic phenotypes but these data are not even shown in the paper for any assay. (This is particularly important given the use of locomotor sensitivity as a proxy for anxiolysis). For example, in fig 3, *Drd2* mice have a hypolocomotive phenotype, which could be responsible for the reduced time in the anxiogenic compartments of the avoidance assays. Is the % distance in these compartments similar to % time? Could it just be that these mice start with very little locomotion (leading to high avoidance score) and thus it is easier to measure a larger % increase in locomotion, driving more locomotion and exploration and lower avoidance score?

6) There is no real explanation or justification for the use of 1.2 vs 2 g/kg alcohol across experiments in the paper. The authors are typically using 2 g/kg for locomotion, calcium activity, etc, but 1.2 g/kg for avoidance behavior, then suggesting these are related. Fig supp 4 shows that there is no alcohol-induced stimulation in locomotion at the lower dose. Even when they run both doses of alcohol, such as in the GCAMP FP in Fig 2, they do not find that 1.2 g/kg has the increased activity effect in dSPNs that 2 g/kg does, yet they then generalize to alcohol increasing this activity, even when interpreting anxiety results at the lower dose. Can the authors justify and clarify the dose used for each experiment and describe why the use of 2 g/kg locomotor stimulation is used as a proxy for the anxiolytic effects of 1.2 g/kg?

7) How is the alcohol stimulation effect segregation performed? It looks like there may be a cap on how much time a mouse will explore the open zones (perhaps before becoming habituated?), and the suggestion that high anxiety mice are more sensitive may just be because they can actually increase. Have the authors tried conditions for these tests that have higher basal avoidance (higher lighting conditions, for example) in these same mice to see if alcohol-insensitive mice display no alcohol-induced increase in exploration in that case?

8) The authors describe using both males and females, however they do not examine how sex may play a role here. There are known differences in basal locomotion, basal anxiety, and the effects of acute alcohol. Are there sex differences in the basal anxiety and magnitude of alcohol-induced anxiolysis (and are there differential sensitivities to alcohol-induced anxiolysis in males and females for different doses of alcohol) here?

9) The D1R and D2R manipulations are different in nature, with the D2R manipulation being a developmental KO that is more broad across the striatum than the D1R mice. How might this affect the results (especially the D2R KO) in terms of D1/D2 balance, etc?

Further, while it is clear that there is an intimate relationship between D2R and D1R, manipulating D2R results in a variety of compensatory changes (especially as a developmental manipulation) beyond increasing D1R expression/function in the DMS. Is there any evidence that local D1R pharmacology is altered in this D2R model? Is there any evidence that D1R and D2R manipulated mice have a correlation between receptor function/availability (degree of knock down/expression, binding, D1/D2 ratio, etc.) and behavioral expression (basal, alcohol-stimulated avoidance)? Why is the ratio different across striatal subregions, and how is that related?

In 4h, it looks like there is a bimodal distribution in the *Drd2*Het mice. What is different about these mice (sex, other behavioral phenotypes, etc)?

DRD2 KO mice often have a less robust or opposite phenotype compare to HETs. What is the explanation for this?

Are there any differences in water consumption between genotypes?

Other issues:

Fig 1 shows 2 g/kg alcohol stimulatory effect (according to methods and results text) but the label says 1.2 g/kg.

The introduction and discussion are very brief. The Introduction does not present any information about the role of the striatum or D1/D2 neurons or signaling to justify the study.

Supp Fig. 1A,E, can the authors indicate the significance in the correlation matrices?

For Fig 2k-o, is there any evidence that the signal is not just decaying over time, or that picking the mice up to inject them is not altering the quality of the signal? is there a saline-saline-saline-control group?

For Fig 1 c and f, the text says this was 2 g/kg but the label on fig 1f says 1.2 g/kg.

The methods describe alcohol CPP but I cannot find any CPP data.

Version 1:

Reviewer comments:

Reviewer #1

(Remarks to the Author)

The authors have answered all of my queries extensively, with additional details and the inclusion of multiple new experiments. I don't have any further questions - it looks great.

Reviewer #2

(Remarks to the Author)

The authors have adequately addressed all of my concerns.

Reviewer #3

(Remarks to the Author)

The revised manuscript from Bocarsly et al. is a refined characterization of how pre-existing anxiety states affect alcohol-related behaviors and physiological effects. Overall, I think the authors addressed the questions and this is a compelling paper. While there are some responses for which I would prefer some direct evidence, I appreciate their careful consideration of the issues I raised and their ability to contextualize these in the broader literature. There are just a few sticking points for me that I think need to be resolved in order to agree with the overall interpretations.

1) I think there is a remaining issue of the effects of sex here. I do realize that they may be underpowered to examine SABV for all of the measures and appreciate the amount of work in this study. However, they are studying phenomena that have known sex differences using mixed sex cohorts, and they say they have resolved that there is no effect of sex, but they only show F separate from M twice, and in one of those cases they show a sex difference. Therefore, I strongly feel that further

consideration of sex be given to some degree and to each facet of the paper, even if they cannot fully resolve this, in order to contextualize the interpretation of their data.

a) In Supp 2D, they say there is no interaction between sex and alcohol. But, is there a main effect of alcohol, and what are the post hoc comparisons within sex on that? Is this effect driven by females? Was this run as a repeated measures ANOVA? Can they connect the dots between saline and alcohol within mouse to see this change? Also, why aren't there any error bars? Why is Supp 2E-I not broken down by sex, as it uses these same data?? The authors state in the response that they focus on assessing M vs F in this cohort because of the high N, but this evaluation is incomplete.

b) They show in Supp Fig 5, but don't really address, that there was no correlation between alcohol depression of evoked DA release and % time in open zones in females (Supp 5E), while the negative relationship between these in males in 3B is a finding contributing to the overall interpretation. It looks like there is still an alcohol-evoked depression of evoked DA in a similar % range as in males, but it seems relevant to show the time course for females as they do in males, and also to show it in the main figure and interpret this result in the text. How should we interpret these other data, knowing that this relationship is sex-dependent? How do other data in Fig 3 look between males and females, as the male, but not female, data for 3b/supp 5e are used to interpret the combined sex data for D1 and D2 function and relationship to behavior? For example, what is the correlation between D1-D2 ratio and open zone time in 3A broken down by sex? Similar sex difference to 3b/supp 5e?

c) Did the authors look at sex effects in other measures of anxiety, locomotion (especially stimulatory effects of alcohol, which have known sex differences), proportion of high vs low response mice, etc, including D1 and D2 manipulations?

2) The authors did do a locomotor assay to show that avoidance behavior was not due to a locomotor effect, which is helpful for their interpretation. But, they did not add the total and % distance traveled, etc for the avoidance assays. These would be helpful for interpreting the anxiety-like behavior.

3) In Supp Fig 2G, they show rep traces of the animals' locomotion. The low risk avoidance alcohol tracking is clearly flawed, as there are a ton of lines across the maze that are impossible. This calls into question the accuracy of these numbers more generally – I understand that EZM can be tricky to track, but this raises a concern.

4) For within-subjects measures, the authors should connect the two data points for each individual mouse with a line so the reader can see the individual mice. They are for some but not others (I think, according to how the methods are described).

5) In Fig 2D-R, which mice are high vs low risk avoiders, and does this affect the gcamp results and effect of alcohol?

6) Why are the data in 4H represented with violin plots without individual data points, as the rest of the figure? Based on violin plots, it looks like the hets have a high outlier that may be skewing the data.

7) 4N – a few data points are extremely high. Is this physiologically possible in mice? and, again there are huge sex differences in basal and stress-induced CORT that muddy these results.

8) 5G – is there any high vs low breakdown for the mutant mice, as for the controls? This result is not really explained in the manuscript.

9) The effects in 5J are somewhat bimodal, and there are known effects of sex on quinine-adulterated drinking. Is there a sex effect here? Are males or females protected from the effect?

Version 2:

Reviewer comments:

Reviewer #3

(Remarks to the Author)

I appreciate the authors seriously considering and addressing all of the remaining questions I had. I think this is a compelling manuscript that offers a lot to the field.

Response to Reviewers comments - NCOMMS-23-20766

We are very grateful to the Reviewers for their insightful and helpful comments on the original manuscript. To address these questions, concerns, and recommendations, we have performed several additional experiments listed below. The revised manuscript includes a whole new figure and has been completely reorganized. Because of your comments and all the work on the revisions, the new manuscript is much stronger and improved. Our most sincere gratitude for your participation in this helpful process.

List of new experiments:

- 1- Repeated EZM measurements in C57BL6/J mice**
- 2- Correlation of EZM performance after saline and Drd1/Drd2 mRNA levels**
- 3- Correlation of EZM performance after alcohol and Drd1/Drd2 mRNA levels**
- 4- Correlation of saline EZM performance and acute alcohol effects in striatum**
- 5- Locomotor response after alcohol 1.2 and 2 g/kg**
- 6- Additional animals run in Intellicages for operant alcohol drinking and quinine adulteration**
- 7- Locomotor tests under familiar conditions in iSPN-Drd2HET and littermate control mice**

Below in bold is a detailed response to each of the reviewer's comments and questions.

Reviewer #1:

The manuscript by Bocarsly et al., Shift of balance in dopamine D1 and D2 receptors enhances the anxiolytic potency of alcohol and promotes punishment-insensitive drinking in mice, attempts to explore the causal relationship between anxiety and alcohol in an inbred mouse model. They show that a low dose of alcohol enhances anxiolytic-like behaviors as compared to saline controlled mice, and that this is in part mediated by striatal D1R-expressing projection neurons. The authors use fiber photometry to demonstrate D1R-expressing neurons are recruited during these behavioral assays. The use of striatal specific Drd1a knockdown mice, and Drd2Het/KO mice, is interesting. While the overall hypothesis is interesting, I don't believe the data adequately tests it and much additional work is needed to support the conclusions.

Major Points

(1) The hypothesis that basal anxiety-like states predicted drinking, or that alcohol provide relief for anxiety-like behaviors, is not tested in this design. The shifts in overall population behavior in Figure 1 do not get at this point. Showing that these behaviors are correlated with each other (Supp Figure 1) also does not answer this. While I appreciate the difficulty of using non-repeatable assays for measuring anxiety-like behaviors, running saline-exposed mice and alcohol-exposed mice in parallel simply does not answer this question. The authors would need to include experiments that actually address this point (some options):

- a. repeatable anxiety assays (eg marble burying) to demonstrate that basal anxiety predicts the response to ethanol
- b. CORT levels at baseline, to show basal anxiety correlated with EtOH-governed responses
- c. Pretest in one assay (eg EZM), sort the mice based on phenotype, and show predictive validity for post-EtOH effects in another assay (not just correlations)

We agree with the Reviewer that correlative analysis is not sufficient evidence for a causal link. In order to identify more direct evidence for a possible causal association, we have followed the recommendations from the Reviewer and performed additional experiments as recommended a and c.

- a) **We performed repeated EZM analysis after saline and alcohol in randomized order and, after thorough validation with saline-saline groups, we found that mice with increased risk aversion after saline indeed show the larger change in the EZM performance after alcohol (Figure 1 I-M).**
- b) **We carried out manipulations of the D1/D2 receptor ratio by creating a transgenic mouse with decreased levels of striatal D2 receptors. These mice showed higher risk avoidance and the largest change in performance after alcohol (Figure 4-5). These experiments show the predictive validity of the D1/D2 receptor ratio for the risk-avoidance phenotype and the alcohol relief.**

We are grateful to the reviewer for suggesting these experiments which have strengthened the conclusions of our study.

(2) Similar to point 1 - a key argument in the abstract is that the anxiolytic effects of alcohol predispose individuals to punishment-resistant drinking. There is not nearly enough evidence in the manuscript to make this claim. The D2R mutants happen to exhibit both alcohol-induced anxiolysis and punishment-resistant drinking phenotypes. To claim that anxiolytic effects make them prone to punishment-resistant drinking, the authors would have to separately show that anxiety-resistant mice generally show punishment sensitive drinking right, and if you manipulate the anxiety phenotype you should be able to manipulate the drinking. Otherwise the authors simply can't say this association is causal.

We agree with the Reviewer that additional experiments were needed to strengthen the causal link and test directly the associations. The revised manuscript includes 5 experiments (3 of which are new) that address this important comment:

- 1) **We carried out a new experiment to directly test whether some wildtype mice had a stronger anxiolytic response to alcohol than others. We tested a large cohort of wildtype mice and found that mice with high risk-avoidance show the largest change in performance after alcohol (Fig. 1J-M).**
- 2) **Alcohol's acute effects on dopamine release in ex vivo striatal slices were also stronger in male mice with high risk-avoidance (Fig. 2B), bridging behavioral performance with ex vivo measurements.**
- 3) **The ratio of dopamine receptor expression in the striatum of wildtype mice was correlated with the risk-avoidance phenotype, the magnitude of the alcohol change and the alcohol drinking preference. Mice with high mRNA ratio for D1:D2 receptors showed high risk-avoidance (Fig. 3A), larger percent change in performance between saline and**

alcohol (Fig. 3C) and higher preference for alcohol drinking in a two-bottle choice task (Fig. 3D).

4) Manipulations of D1:D2 receptor ratio altered alcohol anxiolytic response: lowering the striatal ratio by decreasing *Drd1* expression dampened alcohol anxiolytic effect (Fig. 2B-C) and increasing striatal ratio by decreasing *D2* receptor expression enhanced alcohol anxiolytic effect (Fig. 5A-C).

5) Manipulation of D1:D2 receptor ratio altered the risk-avoidance phenotype and punishment resistant drinking: increasing D1:D2 ration by decreasing *Drd2* expression (Fig. 4A-F) promoted a risk-avoidance phenotype (Fig. 4I-N) and punishment resistant drinking (Fig. 5J-K).

The revised manuscript now offers more direct evidence for a causal link between the ratio of striatal dopamine receptors, risk-aversion phenotype, alcohol anxiolytic efficacy, and alcohol drinking behavior.

(3) Figure 1 uses 1.2 g/kg alcohol, which only modestly recruited striatal D1R neurons (and thus the authors shift to include 2 g/kg with the fiber photometry). It would be interesting to know if 2 g/kg was more predictive in all of their work.

We appreciate the suggestion, and the revised manuscript includes justification for the choice of dose for EZM. The dose of 1.2 g/kg is more appropriate for EZM because it is below threshold for locomotor activation in this mouse strain (Figure 1D) and it is a dose previously used in C57BL6/J mice (Cadwell et al. 2006). It turns out that a higher dose of 2 g/kg is already the peak for locomotor activation which could potentially interfere with the EZM measurement of explore-avoid conflict, and we also lose many more animals at this dose because they fall off the maze (Figure 1D).

(4) I don't believe that C57Bl/6J mice are the appropriate strain for the overall study. These mice are inbred and in reality do not show much range in anxiety-like phenotypes, which is the point of the study.

It is a common belief that genes are the main source of phenotypic variability but plenty of recent studies are broadening our appreciation for other factors that unfold during development and contribute to individual differences in structural brain plasticity and behavior. One of these early studies published in *Science* in 2013 measured the emergence of individuality in genetically identical mice (Freund et al., 2013). There is a long list of published work on this matter and we hope our study joins the list by contributing evidence for phenotypic variability within inbred mice on the explore-avoid conflict tasks and the alcohol response. A recently published study in *Biological Psychiatry* shows individual variability among male C57Bl/6J mice in their response to reward, which was also found to predict future alcohol drinking (Montgomery et al., Biological Psychiatry 2024). Thus, this is a current and relevant topic of study. Our study shows phenotypic variability in risk-avoidance and the response to alcohol, which we found is correlated with striatal expression of *Drd1* and *Drd2* genes in these genetically

identical mice. Our findings suggest other factors, including epigenetic and environmental, rather than genetic differences, account for the individuality. It is expected for phenotypic variability to be even larger in genetically diverse populations. However, under those conditions, isolating the mechanisms could be even more daunting and complex. Inbred mouse lines show some range of phenotypic variability that can be mined to improve our understanding of the mechanism. This is a list of published studies addressing this question using inbred mice:

1. O. Forkosh, S. Karamihalev, S. Roeh, U. Alon, S. Anpilov, C. Touma, *et al.* Identity domains capture individual differences from across the behavioral repertoire. *Nat Neurosci*, 22 (2019), pp. 2023-2028
2. J. Freund, A.M. Brandmaier, L. Lewejohann, I. Kirste, M. Kritzler, A. Krüger, *et al.* Emergence of individuality in genetically identical mice. *Science*, 340 (2013), pp. 756-759
3. D.R. Levy, N. Hunter, S. Lin, E.M. Robinson, W. Gillis, E.B. Conlin, *et al.* Mouse spontaneous behavior reflects individual variation rather than estrous state. *Curr Biol*, 33 (2023), pp. 1358-1364.e4
4. R. Lathe. The individuality of mice. *Genes Brain Behav*, 3 (2004), pp. 317-327
5. S. Stern, C. Kirst, C.I. Bargmann. Neuromodulatory control of long-term behavioral patterns and individuality across development. *Cell*, 171 (2017), pp. 1649-1662.e10
6. N. Torquet, F. Marti, C. Campart, S. Tolu, C. Nguyen, V. Oberto, *et al.* Social interactions impact on the dopaminergic system and drive individuality. *Nat Commun*, 9 (2018), p. 3081
7. B. Juarez, C. Morel, S.M. Ku, Y. Liu, H. Zhang, S. Montgomery, *et al.* Midbrain circuit regulation of individual alcohol drinking behaviors in mice. *Nat Commun*, 81 (2017), p. 2220
8. V. Krishnan, M.H. Han, D.L. Graham, O. Berton, W. Renthal, S.J. Russo, *et al.* Molecular adaptations underlying susceptibility and resistance to social defeat in brain reward regions. *Cell*, 131 (2007), pp. 391-404
9. D. Chaudhury, J.J. Walsh, A.K. Friedman, B. Juarez, S.M. Ku, J.W. Koo, *et al.* Rapid regulation of depression-related behaviours by control of midbrain dopamine neurons *Nature*, 493 (2013), pp. 532-536
10. K. Akiti, I. Tsutsui-Kimura, Y. Xie, A. Mathis, J.E. Markowitz, R. Anyoha, *et al.* Striatal dopamine explains novelty-induced behavioral dynamics and individual variability in threat prediction. *Neuron*, 110 (2022), pp. 3789-3804.e9
11. D.M. Walker, H.M. Cates, Y.E. Loh, I. Purushothaman, A. Ramakrishnan, K.M. Cahill, *et al.* Cocaine self-administration alters transcriptome-wide responses in the brain's reward circuitry *Biol Psychiatry*, 84 (2018), pp. 867-880
12. S. Mondoloni, C. Nguyen, E. Vicq, M. Piscato, J. Jehl, R.D. Durand-de Cuttoli, *et al.* Prolonged nicotine exposure reduces aversion to the drug in mice by altering nicotinic transmission in the interpeduncular nucleus. *eLife*, 12 (2023), Article 80767
13. L. Willmore, C. Cameron, J. Yang, I.B. Witten, A.L. Falkner. Behavioural and dopaminergic signatures of resilience. *Nature*, 611 (2022), pp. 124-132

14. K.B. Leclair, K.L. Chan, M.P. Kaster, L.F. Parise, C.J. Burnett, S.J. Russo. Individual history of winning and hierarchy landscape influence stress susceptibility in mice. *eLife*, 10 (2021), Article e71401
15. L. Li, R. Durand-de Cuttoli, A.V. Aubry, C.J. Burnett, F. Cathomas, L.F. Parise, et al. Social trauma engages lateral septum circuitry to occlude social reward. *Nature*, 613 (2023), pp. 696-703

(5) The authors indicate mice were used ranging from initial PND 45-120. This earlier range is well within adolescence, a known time period of altered response to alcohol and with differing anxiety-like phenotypes. Do the authors see any differences when accounting for age?

Thank you for this comment, which requires an answer and a clarification. The answer is that age is an important variable for EZM tests, as mentioned by the reviewer. We have observed mice become more risk averse as they grow older. For this reason, the age of behavioral testing has been kept consistent and within a narrow range for all behavioral tests in this study. We now state the age range of behavioral testing, rather than age at the start of all the experiments.

Methods reads: “All experiments, unless otherwise stated, used male and female mice (C57BL6/J background, 8-10 weeks old at time of behavioral testing) that were group-housed on 12h:12h light cycle (6:30 on, 18:30 off) with ad libitum access to standard rodent chow and water.”

Six-weeks old mice were only used in the experiments requiring knockout of *Drd1* gene in striatum by EZM testing day (10 weeks old). In this experiment, mice received intracranial injection of Cre recombinase 4 weeks before testing to guarantee a reduction in protein levels. We have revised the range to reflect age at testing.

(6) Figure 1J – wouldn't the better experiment be to show that saline performance on the EZM predicted drinking behavior in the two bottle choice?

We agree that testing this other correlation could be a good addition to Figure 1. We added, however, the repeated EZM experiment with saline and alcohol, as recommended by this Reviewer, and which is a critical test of the main hypothesis of Figure 1. Second, the other findings of the study show that overall alcohol consumption is not significantly altered in the mice with high risk-avoidance, at least not during the first 3 weeks of exposure (Figure 5), but rather the punishment sensitivity of the drinking. Because we found that mice with high risk-avoidance are more sensitive to alcohol relief, the overall drinking might not be expected to be different. For these reasons, we focus on expanding the investigation of the molecular and circuit mechanisms underlying these associations, such as the additional experiments of new Figure 3 that correlate EZM performance and dopamine receptor mRNA levels (new Figure 3).

Reviewer #2 (Remarks to the Author):

I express first my condolences for the sad loss of Patrick Hong and I commend you for honoring them in your manuscript.

In this paper, the investigators are attempting to answer the question on whether anxiety-like phenotypes precede problematic alcohol use by using a mouse model of behavioral responses to alcohol after testing for baseline anxiety-like behaviors. They utilize an impressive multidisciplinary approach that focuses on the expression of dopamine receptors in the striatum. They conclude that the balance of D1 dopamine receptor expression and D2 dopamine receptor expression in different populations of spiny projection neurons (also known as medium spiny neurons, MSNs) accounts for the anxiety-like behavior and the anxiolytic response to alcohol. The study is exciting and important. I have a number of concerns and critiques that I will present below. In the absence of line numbers, I will refer to page numbers and paragraph numbers where I need to comment on a specific statement.

We are grateful for the sympathetic comment about Patrick. We also appreciate that the Reviewer found the study to be exciting and important.

1. The statement (page 2, paragraph 2) that rodents consume more alcohol after a period of abstinence is not necessarily true. There are many studies that show that this is not universally true. In addition, the argument that such a behavior is due to anxiety does not really follow. **We have revised the statement of page 2. We agree with the Reviewer that the evidence currently available in the literature is inconclusive on this regard and thus the need for further exploration that justify the importance of our study. We have made this clearer now.**

2. It was good to include both males and females in the study. It is never made clear though whether there were equal numbers of males and females of every genotype in every experiment. An imbalance in males and females could skew the data. **Thanks for the reminder. We have now included the proportion of the sexes, which was as close to 50% as possible. This was added to the methods section.**

3. Considering that there were many statements about individual variability, I found it surprising that there were very few, if any analyses that specifically determined within-animal correlations. **We carried out additional experiments and now report on several within-animal correlations. Figure 1 now shows new within animal correlations for EZM performance after saline and alcohol to supplement the preexisting within subjects test examining alcohol sensitivity and anxiolytic potency of alcohol. Further, Figure 3 shows all new experiments with within-animal correlations on EZM performance and striatal expression of dopamine receptors and acute alcohol effects on dopamine release in brain slices.**

4. There were multiple instances of referring to correlations of 0.4 to 0.6 as strong correlations. I know this is a bit of a semantics argument, but I would not consider correlations around 0.5 to be strong, but rather moderate.

We agree with the Reviewer and have revised the statements to reflect this.

5. I lost count of how many times there were errors in which panels of which figures were being referred to in the text. Supplemental Figure 3 was never referenced. Supplemental Figure 4 was only ever referenced in the methods. It went straight from Supplemental Figure 2 to Supplemental Figure 5. I am not sure if these data were even discussed.

We apologize for these errors. We had reorganized the figures one last time before submission and we didn't check the references to figures enough.

6. The units for Figure 1A and 1B in the text do not match the units in the Figure itself.

We have fixed this mismatch. Thank you for noticing.

7. It was not clear how the doses of alcohol were chosen and it wasn't considered that individual variability or genotype differences might represent shifts in the dose-response relationships rather than an absence or presence of a response altogether.

The revised manuscript includes the justification for the dose selection for the EZM and locomotor test. We now also include new data on the alcohol effect on locomotion at the two doses (Fig. 1D) that clearly shows 1.2 g/kg alcohol is not sufficient to induce locomotor activation while 2 g/kg alcohol is. A dose response for the EZM task was not possible because increasing the alcohol dose led to animals falling off the maze, likely due to locomotor activation and balance impairments, and this caused the loss of too many data points.

8. Shifts in distribution were provided/described a few times, but there were not analyses to show a significance in these shifts.

We have removed these plots to accommodate the new data collected that includes the repeated EZM and its validation. This new experiment allows for within subject comparison and thus provides more direct evidence for an association between saline and alcohol performance and a second independent measurement.

9. Locomotion data (Figure 1) were provided for the open field, but not for the EZM or the light dark box which was inconsistent.

In the previous version, the locomotor activity was measured in locomotor boxes using beam breaks, not open fields. A new experiment, added to Figure 1, shows that while 2 g/kg alcohol leads to increased locomotor activity, 1.2 g/kg (the dose used on the EZM and light-dark box) does not lead to changes in locomotion. This has all been clarified and added to the revised result section.

10. An increase in alcohol consumption was assessed as a corollary of anxiety-like phenotype, but total fluid consumption was not provided. Polydipsia is an alternative explanation rather than enhanced alcohol consumption.

This is a great idea and suggestion by the Reviewer. We have explored the possibility of polydipsia and analyzed the overall intake and the water consumption as a function of the mice EZM performance. We found no correlation between the water intake and the overall liquid intake as a function of EZM performance. The data is now presented in Figure 1 for control mice and Figure 5 as well as Suppl. Figure 2 for mice with high D1:D2 ratio.

11. What was the rationale for performing photometry in the DMS rather than some other subregion of the striatum (DLS, VMS, ventral striatum)?

We appreciate the question, and we include revisions to address this issue: 1) we expanded the introduction and now include a new paragraph on dorsomedial striatum role in regulating alcohol drinking and the specific plasticity that takes place in D1 striatal projection neurons following chronic exposure. 2) We reorganized the results and figures to show first the finding that D1-like receptor antagonist blocks and D1R knockdown in the dorsomedial striatum attenuates alcohol effects in the EZM. Having the foundation of these findings, the subsequent exploration of D1 striatal projection neuron activity follows more logically.

12. It was not explained what the relevance of the decrease in the AUC of the calcium signal was following alcohol treatment.

This is a puzzling observation that was initially unclear how to interpret it. However recently, we became aware of a published study showing the impact of blood flow on photometry signals (Zhang et al. 2022). Based on the similarities of the signals, we consider that the drop in baseline fluorescence we observed is likely the result of hemodynamic changes caused by alcohol and/or neuronal activation. We now include a statement about this in the result section (page 8):

“We noted that in addition to increased frequency of spontaneous transients, alcohol produced a slow decrease in the signal baseline, quantified in the area-under-the-curve (AUC) measurements (Figure 2Q-R). This change in baseline fluorescence is likely to reflect hemodynamic changes in response to neuronal activation, as recently reported (Zhang et al. 2022).”

13. What is the explanation for why there was only a 55% drop in Drd1a levels with the genetic approach? Why was it not higher? I don't expect 100%, but 55% seems low.

We agree with the Reviewer that this manipulation is a knockdown leading to a partial reduction in Drd1 expression rather than knockout. We think that the biggest factor influencing the mRNA expression is the degree of overlap between the viral infection region and the tissue sample collection for qPCR. The injection site is localized to

dorsomedial striatum and while vector spread is expected, it was not possible to check the degree of infection before the tissue collection for mRNA, which must be done on ice and very quickly to prevent mRNA degradation. Thus, we were unable to check for fluorescence before tissue collecting. This issue is now discussed in the method section (page 24) and reads as “For the *Drd1* knockdown experiments of Figure 2, the viral vector injection was localized to dorsomedial striatum and tissue collected 4-6 weeks later and after behavioral testing. Due to the need for prompt dissection of tissue under RNase free conditions to prevent mRNA degradation, it was not possible to confirm the infection site and spread by assessing fluorescence.”

14. I understand the difficulties of genetic manipulations of D2R expression, but there is a bit of a disconnect between the DMS-focused D1R experiments and the breeding strategy for manipulating D2R expression that produces a loss of D2R striatum-wide. This should be addressed as a limitation of their study.

We agree that this could be described as a limitation of the study. However, we consider that the striatum-wide manipulation of D2 receptor expression has strong face validity for the human condition given the well-known clinical findings that alcohol abuse displays low D2 receptor availability across the striatum. We now included a discussion of this topic in the revised manuscript (page 16):

“...This manipulation produced mice with a two-fold increase in the D1/D2 ratio in the striatum compared to littermate controls. Note that the genetic manipulation employed here produced striatum-wide changes in the D1 to D2 receptor ratio. These findings could have strong clinical validity given the well-known findings that individuals with AUD display low dopamine D2 receptor availability across the striatum...”

15. The D2R expression manipulations all decreased locomotion. How did the investigators disambiguate the differences in time spent in different areas from just reduced motor activity? **This is a valid concern, and we performed an additional experiment to test locomotor activity in the home cages of mice with low D2Rs plus included two other points that argue against a locomotor bias on these explore-avoid conflict tasks:**

- 1) **To disambiguate between locomotion and exploration, we performed locomotor measurement over many days while mice were housed in their homecage. We assessed baseline motor output in iSPN-*Drd2*HET mice and littermate controls and found no difference in locomotor activity when measured in the homecage, indicating no motor impairment in iSPN-*Drd2*HET mice (Figure 4G,H). However, when tested in a novel arena, iSPN-*Drd2*HET mice showed reduced distance traveled, indicating a selective reduction in exploratory behavior, without locomotor impairment (Figure 4I). Furthermore, these experiments indicate that tests performed in novel environments are not just assessment of movement but rather bias towards exploration and motivation to explore.**

- 2) We performed a battery of tests and iSPN-Drd2HET performance was altered compared to control in all of them. We assessed EZM, light and dark box, open field, novelty induced feeding suppression, and corticosterone levels, which is independent of locomotion (Fig. 4).
- 3) Cocaine effect on EZM was another control test performed to disambiguate locomotor stimulation from increased open zone exploration in EZM. Cocaine is a stimulant drug with no reported anxiolytic effects (Zimmerman, 2012). In fact, cocaine increased speed in the EZM but didn't not increase time in the open (Suppl. Figure 3G). Thus, we interpret these findings to reflect that increased locomotion does not always result in more exploration of open zones.

16. The lack of change in the full D2R KO in Figure 3E is ignored.

We apologize that this was not mentioned in the original submission. With the restructuring of the manuscript and the addition of new data, this piece of data on iSPN-Drd2KO is not included and the additional data on iSPN-Drd2KO is now in Suppl. Figure 6.

17. I have major concerns with the interpretation of the quinine consumption experiments, based on how the data were presented. First, the group sizes are very different 6 vs. 13 animals. Only 5 of the 13 mutants maintained drinking in the presence of quinine. The data are all baselined, so there is no way to determine if more alcohol consumption led to greater quinine-resistance or, considering that of the 13 mutant mice there were a lot that consumed very little alcohol, that when the data are normalized, there isn't much a decrease in consumption because they were already drinking very little alcohol. The groups should be better balanced and the data presented in a more clear fashion.

We have performed additional experiments and increased the sample size of both control and iSPN-Drd2HET mice (Figure 5H-K). In addition, the raw data is now also shown in the supplementary figure (Supplementary Figure 7E-I) and the statistical analysis shows that iSPN-Drd2HET mice respond differently to quinine than littermate controls (Figure 5K).

The reviewer brings up an important concern that mice that don't drink much alcohol overall, will not further decrease consumption when paired with quinine. Because of this exact reason, the non-drinker mice are always excluded from the punished data (in this case n=1). The normalized data is shown in Fig. 5J. And now we further complemented this data with the raw intake data before and during adulteration (Suppl. Figure H-I).

18. Page 9, paragraph 4 states that there was an increase in D1:D2 binding ratio specifically in the DMS. That is absolutely not what the data show.

We have revised this statement and have removed the DMS mention.

19. It isn't surprising that there is a change in the D1:D2 ratio when there is a substantial drop in D2 expression. I don't think that the investigators presented data to sufficiently prove that the ratio of the expression of the two receptors is what determines anxiolytic baseline behavior and responses to alcohol. I would ask that they use another method to shift the balance of the two receptors, perhaps by selectively overexpressing one of the two types of receptors in a non-knockout mouse to shift the balance when there is not an absence of the other receptor.

We have added 3 additional experiments which provide further support for the dopamine receptor ratio as one factor influencing the risk-avoidance phenotype at baseline and the relief from alcohol in mice. Furthermore, these experiments were carried out in wildtype C57BL6J mice, without the use of transgenic lines, which adds confidence to the findings. The results show that striatal mRNA levels for the Drd1 and Drd2 ratio, but not each receptor alone, are negatively correlated with saline EZM performance and positively correlated with alcohol EZM performance and drinking preference (new Figure 3).

Last, we would like to point out that the transgenic line used is a single allele deletion of Drd2 gene in only one cell type, so D2R are not absent but rather there is 45% reduction in binding capability in the striatum.

20. Page 11, paragraph 3 proposes a model whereby a "jolt of dopamine" produces the behaviors that are imbalanced because of differing ratios of D1:D2. They performed one experiment with cocaine in a wild type mouse (Figure 1L) to show that this had no effect on anxiolytic behavior. If they want to argue that the imbalance of receptor responses leads to different responses to "jolts" of dopamine release, what better way than to test the imbalance with another test of cocaine?

We agree with the Reviewer that this possible interpretation does not make sense and we have removed it.

21. The final paragraph of the discussion section argues that it is the ratio of D1:D2 in the DMS that produces the behavioral outcomes. In no way do their experiments support a specific role for the DMS in any of the behaviors tested.

We agree with the reviewer and this section was revised to exclude mention of the dorsomedial striatum.

22. The age range at the start of experimentation was very wide. Are there concerns that different ages of mice will lead to different behavioral outcomes?

We appreciate this comment which requires clarification from our part. In the original submission we included the age at the start of the experiments but the age that is relevant is the age at the time of behavioral testing. The age at the time of behavioral testing was kept consistent and within a narrow range (8-10 weeks old) for all the experiments in this study, because it does affect performance in EZM tests. We have observed mice become more risk averse as they grow older.

The revised manuscript states the age range of behavioral testing, rather than age at the start of all the experiments, in the method section.

Younger animals (6 weeks old) were only used at the start of the experiments requiring knockout of Drd1 gene in striatum by age of testing in the EZM (~10 weeks old). The reason for this is that reduction in D1R protein and function requires several weeks (~4 weeks). We have revised the method to state age range at testing.

23. There is no statement regarding whether the data were tested for normality before proceeding to use tests that are for normally-distributed data.

This has been added to the statistical section in the methods.

24. The supplemental checklist indicates that blinding was used. For the average reader who will not look at the checklist, it would be good to include this statement in the methods section.

Thanks for this reminder. The revised manuscript now incorporates a statement detailing the experimenter's blindness to the genotype or experimental condition during behavioral testing and analysis.

25. What does Supplemental Figure 5B show that Figure 5C does not already show? Individual values?

We agree there was redundancy in these plots and the intention was indeed to show the individual animal values. Figure 5 was fully revised to incorporate new data.

26. I found the organization/presentation of Figure 1I to be very confusing.

We appreciate this comment and we have revised the whole Figure 1, and specially Figure 1I to make it clearer.

Reviewer #3 (Remarks to the Author):

The manuscript by Bocarsly et al. is an intriguing examination of the roles of D1R and D2R in the dorsomedial striatum, and the balance between them, on the acute effects of alcohol on avoidance behavior and locomotion. This is an area of investigation in the alcohol research field that has been under-explored in the literature, and there is potential for this work to contribute significantly to the field. However, there is some lack of clarity in the presentation and reporting that make it difficult to understand the actual results and evaluate the interpretation of the data.

One broad issue is that there is no evidence that anxiolytic effects of alcohol (and manipulations altering these) are not due to locomotion. This stems from several related questions/issues included below.

We have carried out 7 additional experiments outlined at the top of this letter and have reorganized the manuscript and figures to show that the alcohol dose used for risk-avoidance testing does not produce locomotor activation. The additional data also clarifies the difference between locomotor output and novel environment exploration. The revised manuscript addressed the possible confound of locomotion and the reorganization of figures also improves the logical flow of the study.

1) Lack of clarity about the specific experimental conditions within each figure/panel and inconsistency across them. Was all of the behavior done on the same mice or different groups of mice? Can the authors provide experimental timelines at the top of each fig/relevant group of panels?

We appreciate this helpful comment. We have added a timeline for each experimental procedure.

2) What is the point of the distribution graphs for avoidance behavior, given that the distributions are fairly obvious in the bar graphs above them? They generally look like the gaussian distribution shifts, with typical large variability within group and a high degree in overlap of distribution across groups, suggesting a general effect across mice, however the authors argue that some mice are more susceptible than others to alcohol-ON anxiolysis.

We agree with this observation, and we have removed the distribution plots and have added two additional experiments that are key on directly testing the correlation between the EZM performance at baseline and after alcohol (Fig 1 J-M) and also provide data on alcohol induced exploration/locomotion (Fig, 1D).

3) What is the alcohol stimulatory score? Is it a difference in locomotion between saline and alcohol within each mouse or just the alcohol ON locomotion? If the latter, this should not be referred to as the stimulatory effect (implying within-mouse) because that is not what it is showing. Similarly, the authors frequently refer to the anxiolytic effect of alcohol as the alcohol-ON avoidance behavior, but they generally do not show that it is anxiolytic within-mouse (except Supp Fig 4).

The reviewer is correct, and the stimulatory score was the difference between the saline and the alcohol for each mouse. It was a confusing term that we removed in the revised manuscript. Instead, we directly show the individual difference in stimulation and the response to alcohol and saline in the EZM (Fig. 1N-Q). Furthermore, to address the second part of the comment, we have carried out additional experiments testing the anxiolytic effects of alcohol within mice using two repeated EZM measures, one with saline and the other with alcohol in randomized order. The approach was validated with repeated saline tests. All this new data is shown in the revised Figure 1H-M.

4) Relatedly, why is the alcohol stimulatory score used as a proxy for the anxiolytic effects? Why isn't an anxiolytic score as in Supp Fig 4 used throughout to more directly show alcohol-induced anxiolysis?

We apologize the data presentation was so confusing and we have revised and clarified. The answer to the question is: No, the stimulatory effects were not used as a proxy for the anxiolytic effects but rather we found an association. We understand that the revealed association raises concern that alcohol effects are just producing movement. We have done multiple additional experiments to address this possible concern and provide evidence that alcohol is promoting exploration and risk-taking behaviors, rather than movement per se. Please see a more extensive response on disambiguating anxiolytic and locomotion effects on question 5.

4b) This issue permeates the other analyses and data interpretation. For example, they use the alcohol stimulatory score in correlations with other measures like basal avoidance and interpret that as high basal avoidance being related to alcohol-induced anxiolysis. Is there any evidence for this relationship that does not depend on the locomotion? For example, is alcohol ON avoidance correlated with basal avoidance? Same for alcohol drinking phenotype (as in 1J)-- how does alcohol drinking compare to basal (saline) anxiety?

These are important questions and we have performed repeated EZM experiments to assess the association directly. The within mouse design experiments shows that mice with high risk-avoidance find more relief from alcohol. We also found a positive correlation between alcohol relief in the EZM and alcohol consumption in the two-bottle choice paradigm, while water drinking was not correlated (Fig. 1E-G).

For more experiments on disambiguating anxiolytic and locomotion effects, please see response to question 5 below.

5) The authors do not seem to present the locomotor data from the anxiety assays, nor examine the relationship between that locomotion and avoidance behavior on tests. As there is often a locomotor phenotype at baseline or in response to alcohol, differences in locomotion at basal/alcohol conditions may be a confound for interpretation of anxiolytic phenotypes but these data are not even shown in the paper for any assay. (This is particularly important given the use of locomotor sensitivity as a proxy for anxiolysis). For example, in fig 3, Drd2 mice have a hypolocomotive phenotype, which could be responsible for the reduced time in the anxiogenic compartments of the avoidance assays. Is the % distance in these compartments similar to % time? Could it just be that these mice start with very little locomotion (leading to high avoidance score) and thus it is easier to measure a larger % increase in locomotion, driving more locomotion and exploration and lower avoidance score?

This is a valid concern, and we performed an additional experiment to test locomotor activity in the home cages of mice with low D2R plus included two other points that argue against a plain locomotor bias on these explore-avoid conflict tasks:

- 1) To disambiguate between locomotion and exploration, we performed locomotor measurement over many days while mice were housed in their homecage. We assessed baseline motor output in iSPN-Drd2HET mice and littermate controls and found no difference in locomotor activity when measured in the homecage,**

indicating no motor impairment in iSPN-Drd2HET mice (Figure 4G,H). However, when tested in a novel arena, iSPN-Drd2HET mice showed reduced distance traveled, indicating a selective reduction in exploratory behavior, without locomotor impairment (Figure 4I). Furthermore, these experiments indicate that tests performed in novel environments are not just assessment of movement but rather bias towards exploration and motivation to explore.

- 2) We performed a battery of tests and iSPN-Drd2HET performance was altered compared to control in all of them. We assessed EZM, light and dark task, open field, novelty induced feeding suppression and cortisol levels, which is independent of locomotion (Fig. 4).
- 3) Cocaine effect on EZM was another control test performed to disambiguate locomotor stimulation from increased open zone exploration in EZM. Cocaine is a stimulant drug with no reported anxiolytic effects (Zimmerman, 2012). In fact, cocaine increased speed in the EZM but didn't not increase time in the open (Suppl. Figure 3G). Thus, we interpret these findings to reflect that increased locomotion does not always result in more exploration of open zones.
- 4) We measured a battery of tests and HET performance was altered compared to control in all of them. We assessed EZM, light and dark task, open field, novelty induced feeding suppression and cortisol levels, which is independent of locomotion.
- 5) Another way to disambiguate the changes in locomotion and EZM performance is the test done with cocaine administration. This stimulant drug increased speed in the EZM but didn't not increase time in the open (Suppl. Figure 3G). Thus, we interpret these findings to reflect that more locomotion does always result in more exploration of open zones, which is also in agreement with lack of reports on cocaine anxiolytic effects. These findings are also
- 6) There is no real explanation or justification for the use of 1.2 vs 2 g/kg alcohol across experiments in the paper. The authors are typically using 2 g/kg for locomotion, calcium activity,

etc, but 1.2 g/kg for avoidance behavior, then suggesting these are related. Fig supp 4 shows that there is no alcohol-induced stimulation in locomotion at the lower dose. Even when they run both doses of alcohol, such as in the GCAMP FP in Fig 2, they do not find that 1.2 g/kg has the increased activity effect in dSPNs that 2 g/kg does, yet they then generalize to alcohol increasing this activity, even when interpreting anxiety results at the lower dose. Can the authors justify and clarify the dose used for each experiment and describe why the use of 2 g/kg locomotor stimulation is used as a proxy for the anxiolytic effects of 1.2 g/kg?

We justify and revised manuscript to clarify the alcohol dose used for EZM. In brief, we used 1.2 g/kg for EZM because this is the dose used previously in C57BL6/J mice (Cadwell et al. 2006) and was validated early on in our lab on a small cohort of mice. This dose is below threshold for locomotor activation in this mouse strain, now shown in Figure 1D. A higher dose of 2 g/kg shows an increase in locomotor activity (Figure 1D), which could potentially interfere with the EZM measurement of explore-avoid conflict. In initial experiments carried out in our lab for this study were test 2 g/kg alcohol in the EZM and found that we lost data from many mice because they fell off the maze.

Regarding the use of stimulation as a proxy for anxiolytic effects, we apologize again if we misled the Reviewer and we have revised the manuscript to clarify. The stimulatory effects were not used as a proxy for the anxiolytic effects but rather we found an association between them. Please see a more extensive response on this matter in point 4 and 5.

7) How is the alcohol stimulation effect segregation performed? It looks like there may be a cap on how much time a mouse will explore the open zones (perhaps before becoming habituated?), and the suggestion that high anxiety mice are more sensitive may just be because they can increase. Have the authors tried conditions for these tests that have higher basal avoidance (higher lighting conditions, for example) in these same mice to see if alcohol-insensitive mice display no alcohol-induced increase in exploration in that case?

We used both median splits (Fig. 1L) and mean splits for segregation (Suppl. Fig. 2E). Both show similar results: mice with lower exploration at baseline changed the most. We agree with the Reviewer's interpretation that some individuals show a higher degree of exploration at saline level so alcohol doesn't have much of an effect on them because they are already exploring the risky arena. This interpretation further suggests that alcohol promotes exploration by "removing a brake" and that the stronger that brake, the stronger the alcohol effect.

Regarding the suggestion to manipulate risk avoidance behavior in wildtype mice, we are doing a follow-up study investigating the impact of early life stress on risk avoidance phenotype and alcohol relief and drinking.

8) The authors describe using both males and females, however they do not examine how sex may play a role here. There are known differences in basal locomotion, basal anxiety, and the effects of acute alcohol. Are there sex differences in the basal anxiety and magnitude of alcohol-

induced anxiolysis (and are there differential sensitivities to alcohol-induced anxiolysis in males and females for different doses of alcohol) here?

Thanks for this suggestion. To test for possible sex difference in the risk avoidance and alcohol relief, we sorted the data from repeated EZM for which we have a large sample size of each sex and found no difference between males and females in neither the baseline phenotype nor alcohol response. This analysis is now shown in Suppl. Figure 2B.

9a) The D1R and D2R manipulations are different in nature, with the D2R manipulation being a developmental KO that is more broad across the striatum than the D1R mice. How might this affect the results (especially the D2R KO) in terms of D1/D2 balance, etc?

We agree with the Reviewer that the temporal aspect of the dopamine receptor modulation is likely to be a very important factor. We hypothesized that dopamine receptor levels could be modulated during development and in adulthood by both internal factors (e.g. genetic and epigenetic) and external factors (e.g. environmental factors, such as stressor). We hypothesize that modulation of the ratio during adulthood might have milder implications for the overall circuitry. In contrast, embryonic and early manipulations, could have a profound long-lasting impact on brain wiring, specially during early development and adolescence stages. We speculate that differences in brain wiring among individuals are associated with phenotypic variation in risk-avoidance and alcohol relief potency . Inspired by this comment from the Reviewer, the revised manuscript includes this topic in the discussion (page 16):

“We hypothesized that the ratio of dopamine receptors in the striatum may be modulated by both internal factors (e.g. genetic and epigenetic) and external factors (e.g. environmental factors, such as stressor). The age of onset is also likely to be an important factor, with changes to the ratio during adulthood having milder implications for the overall brain wiring and embryonic or early-age alterations of ratio having a more profound, long-lasting impact on brain wiring. We speculate that differences in brain wiring among individuals could be associated with phenotypic variation in risk-avoidance and alcohol relief potency.”

9b) Further, while it is clear that there is an intimate relationship between D2R and D1R, manipulating D2R results in a variety of compensatory changes (especially as a developmental manipulation) beyond increasing D1R expression/function in the DMS. Is there any evidence that local D1R pharmacology is altered in this D2R model? Is there any evidence that D1R and D2R manipulated mice have a correlation between receptor function/availability (degree of knock down/expression, binding, D1/D2 ratio, etc.) and behavioral expression (basal, alcohol-stimulated avoidance)? Why is the ratio different across striatal subregions, and how is that related?

The reviewer raised several very important questions. With regard to the D1R behavioral response, yes, we have carefully validated and studied the D2R knockdown model, and we have shown functional reduction of D2R and increased functionality/response to D1-

like agonists. We now include the citations of these published studies (Lemos, et al., Neuron 2016; Dobbs et al. NPP 2019; Bocarsly et al., Cell Reports 2019).

With regard to the question of why the receptor ratio is different across striatal subregions and how this relates to our work, we can only speculate. We remain very curious about the revelation of regional variations in receptor ratio. To our knowledge, there is no previous notice of this regional difference that is generated because the gradients in D2R and D1R levels across the striatum are not exactly matched. It is possible that the DMS having the highest ratio is an insignificant fact for physiology and behavior; however, given the findings of this study, we hypothesize that this regional difference is related to the unique functions of the DMS, relative to DLS and NAc, and suggest possibly differential regulation by dopamine.

In 4h, it looks like there is a bimodal distribution in the Drd2Het mice. What is different about these mice (sex, other behavioral phenotypes, etc)?

We appreciate the suggestion to look further in it. There is no obvious difference between the animals on the top and bottom clusters of this data set. We have looked at sex differences in EZM performance and the data is now shown in Suppl. Figure 2.

DRD2 KO mice often have a less robust or opposite phenotype compared to HETs. What is the explanation for this?

By studying all the data presented in Figures 4 and 5, we observe that Drd2KO have a stronger phenotype than HET in some measurements and behaviors (novel arena exploration and alcohol stimulation), but for other behaviors, HET and KO are similar (EZM, Light and dark, serum cortisol). We think that the presence or not of a difference in the phenotypes of the KO and HET is informative on the degree of D2R mediate modulation of the specific behavior. It appears that D2R function in striatal projection neurons has a dominant effect and a single allele removal and consequently small reduction in D2R function is sufficient to alter risk aversion phenotypes.

Are there any differences in water consumption between genotypes?

There is no difference in water consumption across genotypes. The data is now included in Suppl. Figure 7.

Other issues:

Fig 1 shows 2 g/kg alcohol stimulatory effect (according to methods and results text) but the label says 1.2 g/kg.

This topic is now clarified. Anxiolytic effects on EZM were assessed with 1.2 g/kg because this dose does not induce locomotor activation, which confounds the interpretation of the results as well as the data collection because more mice fell off the maze at 2 g/kg dose. Alcohol stimulant effects were assessed with 2 g/kg.

The introduction and discussion are very brief. The Introduction does not present any information about the role of the striatum or D1/D2 neurons or signaling to justify the study.

This is a very helpful comment. Introduction was expanded to include background on the important roles of striatal D1/D2 neurons and receptors.

Supp Fig. 1A,E, can the authors indicate the significance in the correlation matrices?

Great suggestion. We have added the p values to the correlation matrices.

For Fig 2k-o, is there any evidence that the signal is not just decaying over time, or that picking the mice up to inject them is not altering the quality of the signal? is there a saline-saline-saline-control group?

We appreciate this suggestion. Yes, in fact we have a saline control, and we didn't observe this decay in fluorescence (see Figure 1Q). Since the original submission, there are now published reports that transient changes in blood flow within the tissue can cause drops in GCaMP fluorescence measured with photometry (Zhang et al. 2022). Based on the similarities of the signals, we consider that the drop in baseline fluorescence we observed is likely the result of hemodynamic changes caused by alcohol and/or neuronal activation. We now include a statement about this in the result section (page 8):

“We noted that in addition to increased frequency of spontaneous transients, alcohol produced a slow decrease in the signal baseline, quantified in the area-under-the-curve (AUC) measurements (Figure 2Q-R). This change in baseline fluorescence is likely to reflect hemodynamic changes in response to neuronal activation, as recently reported (Zhang et al. 2022).”

For Fig 1 c and f, the text says this was 2 g/kg but the label on fig 1f says 1.2 g/kg.

We have corrected this mismatch. Thank you.

The methods describe alcohol CPP but I cannot find any CPP data.

Sorry about this. This section was removed.

RESPONSE TO REVIEWER COMMENTS
NCOMMS-23-20766A-Z
Bocarsly et al.

Reviewer #1:

The authors have answered all of my queries extensively, with additional details and the inclusion of multiple new experiments. I don't have any further questions - it looks great.

Reviewer #2:

The authors have adequately addressed all of my concerns.

We are grateful to Reviewers #1 and #2 for all their previous comments that greatly improved the manuscript.

Reviewer #3:

The revised manuscript from Bocarsly et al. is a refined characterization of how pre-existing anxiety states affect alcohol-related behaviors and physiological effects. Overall, I think the authors addressed the questions and this is a compelling paper. While there are some responses for which I would prefer some direct evidence, I appreciate their careful consideration of the issues I raised and their ability to contextualize these in the broader literature. There are just a few sticking points for me that I think need to be resolved in order to agree with the overall interpretations.

1) I think there is a remaining issue of the effects of sex here. I do realize that they may be underpowered to examine SABV for all of the measures and appreciate the amount of work in this study. However, they are studying phenomena that have known sex differences using mixed sex cohorts, and they say they have resolved that there is no effect of sex, but they only show F separate from M twice, and in one of those cases they show a sex difference. Therefore, I strongly feel that further consideration of sex be given to some degree and to each facet of the paper, even if they cannot fully resolve this, in order to contextualize the interpretation of their data.

We appreciate and value these comments about sex differences. In response, the revised manuscript includes a complete analysis by sex of the multiple experiments of the study. A summary is below:

- a) **Revised Supplementary Figure 2 now includes a complete analysis of the EZM and alcohol data sorted by sex (panels C-J)**
- b) **Revised Figure 3 now shows the slice recording data from females (previously in Supp. Fig. 5) and the time course of the acute alcohol response in both female and male brain slices, as requested by the reviewer.**
- c) **Revised Supplementary Figure 5 shows the striatal expression for the ratio of Drd1 to Drd2, sorted by sex. Significant correlations were identified in both sexes between the receptor expression ratio and the EZM performance after saline and the percent change after alcohol (panels A-B and E-F).**
- d) **Revised result section incorporates the above additions and interprets the sex analysis in the context of the rest of the findings (pages 4,5,8,9).**

1a) In Supp 2D, they say there is no interaction between sex and alcohol. But, is there a main effect of alcohol, and what are the post hoc comparisons within sex on that? Is this effect driven by females? Was this run as a repeated measures ANOVA? Can they connect the dots between saline and alcohol within mouse to see this change? Also, why aren't there any error bars?

Supp. Fig. 2D has been revised and now we report on the full statistical analysis in the figure legend. We also connected the dots corresponding to the same animals in Supp. Fig. 2D, and every plot throughout the manuscript where there are repeated measures. Error bars were added to the bars representing the mean, but they are obscured by the individual data points. We appreciate this helpful comment that has further improved data presentation.

1b) Why is Supp 2E-I not broken down by sex, as it uses these same data?? The authors state in the response that they focus on assessing M vs F in this cohort because of the high N, but this evaluation is incomplete.

This is a good point. We performed complete analysis by sex of this data set. The revised Supplementary Figure 2 now shows data sorted by sex in panels 2E through 2I and includes a new plot, 2J.

1c) They show in Supp Fig 5, but don't really address, that there was no correlation between alcohol depression of evoked DA release and % time in open zones in females (Supp 5E), while the negative relationship between these in males in 3B is a finding contributing to the overall interpretation. It looks like there is still an alcohol-evoked depression of evoked DA in a similar % range as in males, but it seems relevant to show the time course for females as they do in males, and also to show it in the main figure and interpret this result in the text. How should we interpret these other data, knowing that this relationship is sex-dependent? How do other data in Fig 3 look between males and females, as the male, but not female, data for 3b/supp 5e are used to interpret the combined sex data for D1 and D2 function and relationship to behavior? For example, what is the correlation between D1-D2 ratio and open zone time in 3A broken down by sex? Similar sex difference to 3b/supp 5e?

We agree with the Reviewer that these findings are important and relevant. As suggested, we have moved the data for the female mice to the main figure and added the time course of the alcohol effect in both male and female mice to the main figure. As now shown in Figure 3B, alcohol also affects dopamine signals in striatal brain slices from female mice, but the effect is smaller (2WRM ANOVA, significant alcohol x sex interaction $F(14, 420) = 3.502, p < 0.0001$). Additionally, there was no significant correlation between the alcohol effect and the baseline performance in the EZM for the females.

These sex differences in the *in vitro* alcohol effects on evoked striatal dopamine are currently the primary differences identified between the sexes. This difference resembles a recently reported difference in acute effects of alcohol in the amygdala (Munshi et al., 2023), where the brain circuitry of female mice was found to be less sensitive to alcohol than male mice. In the current study, the lack of correlation with behavior may reflect the blunted alcohol effect, which impairs the ability to detect an existing correlation. Alternatively, alcohol's anxiolytic effects may be mediated via different mechanisms in female mice compared to males, as suggested by Munshi and colleagues.

The result section has been revised to incorporate the new analysis with the following statement (page 9) “Using fast-scan cyclic voltammetry in brain slices, we found that the magnitude of the acute alcohol effect was variable (Supplemental Figure 5H) and correlated with EZM performance in male mice. Males that spent less time in the open zones showed a stronger *in vitro* alcohol effect (Figure 3B). No correlation was observed in tissue from female mice and the same concentration of alcohol (80 mM) had a smaller overall effect on evoked dopamine signals in the striatal slices from female mice compared to those from male mice (Figure 3B; Suppl Figure 5G). This is one of the few analyses in the current study that revealed a sex difference. In agreement with recent findings in the amygdala⁴³, these findings suggest that the female brain circuitry is less sensitive to alcohol, and could explain the higher levels of alcohol consumption observed in females compared to male mice.”

As the reviewer suggests that these results lead to the questioning of sex-effect on other data sets in the manuscript, we also have presented sex-split data for other figures, such as Figure 3A (new data in Supplementary Figure 2 and 5).

1d) Did the authors look at sex effects in other measures of anxiety, locomotion (especially stimulatory effects of alcohol, which have known sex differences), proportion of high vs low response mice, etc, including D1 and D2 manipulations?

Both sexes at this age are similarly represented below and above the mean of the saline EZM performance (Supp. Figure 2G). For males, there are 10 above and 13 below mean (43% and 57%) and for females 9 above and 13 below (40% and 60%). These values are not significantly different between the sexes nor from 50% ($p = 0.67$ for males and $p = 0.41$ for females, binomial test Wilson/Brown calculation). Similar findings are replicated in the EZM tests with repeated saline injections (Suppl. Fig. 2J).

We identified sex differences in the effect of alcohol on DA signals in the dorsomedial striatum, with females being less sensitive to the acute alcohol effect in the slice recordings. As requested by the reviewer, we now show both the male and female data, including the time course and the correlation (or lack of) with the EZM performance.

2) The authors did do a locomotor assay to show that avoidance behavior was not due to a locomotor effect, which is helpful for their interpretation. But, they did not add the total and % distance traveled, etc for the avoidance assays. These would be helpful for interpreting the anxiety-like behavior.

We agree with the reviewer on the importance of dissociating the risk avoidance from the locomotor effects. It is also important to note that most common tests for locomotion also assess exploration, which is not just the ability to move, but is influenced by the motivation to seek novelty. We took a multi-pronged approach to disentangling risk aversion from locomotor activity by testing a stimulant substance without anxiolytic properties (cocaine) in the same arena (EZM) that we had tested alcohol, and by assessing the movement in the home cage for the mice with low D2Rs.

We would have liked to also report distance traveled during the EZM, but as the reviewer well noted, the video tracking often lost track of mouse position while in the closed zones, which produced large artifacts in the total distance traveled. Note that this

technical issue does not affect the total time spent in each zone, which was verified by the analyzer. We have revised the method section to include a direct mention of this issue (page 17).

3) In Supp Fig 2G, they show rep traces of the animals' locomotion. The low risk avoidance alcohol tracking is clearly flawed, as there a ton of lines across the maze that are impossible. This calls into question the accuracy of these numbers more generally – I understand that EZM can be tricky to track, but this raises a concern.

We understand the reviewer's concern here and we want to reassure the reviewer of the validity of the data. The images show the distance traces, but the data analysis was performed on the time spent in the open zone (where the mouse can be clearly visualized). The time spent in the open zones was analyzed first automatically by the software and then always verified by experimenters. Thus, the time spent in each location is accurate and validated. The traces of the distance traveled sometimes show the erroneous lines across the maze, when the mouse is in the closed zones, which is why we don't report or trust these calculations of the distance traveled in the EZM. Briefly, this error occurs when the software loses the mouse location as it enters the close arms. This caveat is now addressed in the method section (page 17) with the following statement **"The semiautomatic analysis of time spent in open zones was checked by blind experimenters and found to be accurate most of the time. However, the mouse location while in the close zones was not accurately assessed from the video in most cases, which precluded any measures of total distance travel in the EZM."**

We would also like to note that the traces in question have been removed from supplemental figure 2 in order to make room for the sex-differences that are now presented there.

4) For within-subjects measures, the authors should connect the two data points for each individual mouse with a line so the reader can see the individual mice. They are for some but not others (I think, according to how the methods are described).

Thank you for pointing this out. We have connected the data points for the within-sample comparisons for all possible graphs: panels 1L, 2P, S2F, S2G, S2H, S2J. In Sup. Fig. 2D and E, the graphed comparison will not allow clear connection, so we did not add connecting lines, but these data are the same data graphed in Sup Fig 2E-G with connecting lines.

5) In Fig 2D-R, which mice are high vs low risk avoiders, and does this affect the gcamp results and effect of alcohol?

There is individual variability in the EZM performance of these mice, but it was not correlated to the gCaMP measures. Note that the EZM performance of mice implanted with fiber optic and expressing gCaMP6 in D1R expressing SPNs showed similar mean time in open as the large cohorts shown in Figure 1 (~ 180 sec for 600 s test), indicating that neither the surgery nor the tether itself affected EZM performance.

While the sample size of the cohort of implanted mice was smaller (n=6), we still noted individual differences in the performance - one mouse showing close to 50% of the time spent in the open (Fig. 2J). While the small “n” value makes it possible that we might have missed a correlation between gCaMP measure and time in the open, we interpret these findings, along with the other results of Figure 2, to indicate that D1R expressing neurons are activated during the transition from close to open zones and that D1-SPN activity is higher in the open than close zones. We speculate that this activation will happen in mice with both high and low risk-aversion during the transition into the risky, open zones. The main difference among these groups is how often the risky behavior occurs. The data also shows that alcohol promotes both D1-SPN activation and the transitions to the risky, open zones.

6) Why are the data in 4H represented with violin plots without individual data points, as the rest of the figure? Based on violin plots, it looks like the hets have a high outlier that may be skewing the data.

We appreciate the helpful suggestion. For consistency, we now plot this data as mean \pm SEM. We also performed additional statistical analysis on the distribution of the data and found data from both iSPN-Drd2HET and littermate controls passed all tests for normality and lognormality distributions (D’Agostino & Pearson test, $K2 = 1.778$ for iSPN-Drd2HET and 3.305 for control, $p=0.19$, $n=30$; also passed normality test for Anderson-Darling, Shapiro-Wilk, and Kolmogorov Smirnov).

7) 4N – a few data points are extremely high. Is this physiologically possible in mice? and, again there are huge sex differences in basal and stress-induced CORT that muddy these results.

It is very interesting that, while a majority of the iSPN-Drd2HET mice trend higher than the littermate controls, there is a subgroup of mice that have very high corticosterone levels. While high, these levels are physiologically relevant and have been reported in stress-sensitive transgenic mice (McGill, Bundle, Yaylaoglu, Carson, Thaller, and Zoghbi, PNAS, 2006; Gong, Miao, Jiao, Sun, Li, Lin, Luo, and Tan, PlosOne 2015; Mizobe, Kishihara, Ezz-Din El-Naggar, Madkour, Kubo, et al., 1997, J Neuroimmunol). This bimodal distribution of the data is not sex dependent, and interestingly, can also be seen across behavioral testing in these mice too. While beyond the scope of this publication, we have been questioning what leads to this recurring bimodal distribution.

8) 5G – is there any high vs low breakdown for the mutant mice, as for the controls? This result is not really explained in the manuscript.

Only data from wildtype mice was sorted based on their behavior. For the experiment using genetic manipulation of D2R levels, we focused on the comparison of low D2R to littermate controls in order to test the causality of the manipulation. The aim was to test whether low D2R mice look similar to the high-stimulation wildtype mice. We consider that sorting these mice would deter from this goal. This has been clarified in the text.

9) The effects in 5J are somewhat bimodal, and there are known effects of sex on quinine-adulterated drinking. Is there a sex effect here? Are males or females protected from the effect?

We looked further into the sex composition of the alcohol adulteration experiment. Both sexes are represented in the adulteration-resistant group: 43% of HET mice (6 out of 14)

did not reduce drinking upon adulteration with 0.5 mM quinine, 66% were male and 33% females (4 male, 2 females). For the littermate control mice, only 9% of the mice that continued drinking alcohol despite adulteration (1 out of 11) and this single mouse was male. This information has been added to the result section (page 13).

Regarding the second question, females are represented in the adulteration insensitive group in the HET mice, but at a lower proportion than males. Females insensitive to adulteration were 25% of the total females, while 48% for the males. Thus, the distributions appeared skewed but the low probabilities plus the small sample size preclude us from statistically valid assessments.